# Provably Protecting Fine-Tuned LLMs from Training Data Extraction while Preserving Utility

**Tom Segal** [1]    **Yuval Elovici** [1]    **Asaf Shabtai** [1]

## Abstract

Fine-tuning large language models (LLMs) on sensitive datasets raises privacy concerns, as training data extraction (TDE) attacks can expose highly confidential information. Existing defenses against such attacks either lack formal privacy guarantees or incur substantial utility degradation. We observe that fine-tuning induces widespread probability shifts, yet preserving only a small subset of influential token-level deviations is sufficient; the remaining shifts can be aggressively smoothed with minimal impact on utility. Motivated by this insight, we propose SCP-$\Delta_r$, a Near Access Freeness (NAF)-based algorithm that operates on relative probabilities and explicitly smooths low-impact tokens using a base model. SCP-$\Delta_r$ achieves orders-of-magnitude better theoretical bounds than existing NAF based methods and provides strong empirical protection against TDE attacks with minimal performance loss.

**Code:** https://github.com/ppo1/scp_dr

## 1. Introduction

Large language models (LLMs) are known to memorize parts of their training data, enabling training data extraction (TDE) attacks that recover verbatim sequences from black-box access to a model. Such attacks have been demonstrated against commercial models such as ChatGPT and Gemini (Carlini et al., 2021; Nasr et al., 2025). This risk is amplified by fine-tuning, which is routinely performed on small, high-value, and often private datasets (Patil & Gudivada, 2024; J et al., 2024).
Existing defenses face a fundamental trade-off. Empirical mitigations such as deduplication or modified losses (Kand-

[1]Department of Software and Information Systems Engineering, Ben-Gurion University of the Negev, Beer Sheva, Israel. Correspondence to: Tom Segal <tomsega@post.bgu.ac.il>.

*Proceedings of the $43^{rd}$ International Conference on Machine Learning*, Seoul, South Korea. PMLR 306, 2026. Copyright 2026 by the author(s).

pal et al., 2022; Hans et al., 2024) can reduce memorization in practice but offer no formal guarantees and are increasingly circumvented by new attacks (Nasr et al., 2025). Conversely, differential privacy (DP) provides rigorous protection but often incurs substantial utility loss or computational overhead when applied to LLM fine-tuning (Blanco-Justicia et al., 2022; Du et al., 2025). As a result, there is currently no practical defense with meaningful theoretical guarantees against TDE for fine-tuned LLMs.

In this work, we bridge this gap by developing a theoretically grounded yet practical protection mechanism based on Near Access Freeness (NAF) (Vyas et al., 2023). NAF constructs a model that is provably close, under a divergence measure, to multiple constituent models, guaranteeing safety if at least one constituent is safe for a given input. While NAF has been applied to copyright protection, access control, and privacy (Li et al., 2024; Kalai et al., 2025; Segal et al., 2025), its implications for TDE have not been previously studied. In particular, we show that the widely used NAF based CP-$\Delta$ algorithm can yield weak guarantees in realistic fine-tuning settings, since memorization induces sharp probability spikes on a small number of tokens.

**Sparse relative probability shifts.** Our key observation is that fine-tuning concentrates the most significant changes in model outputs on a relatively small subset of tokens, even though absolute token probabilities may vary widely due to normalization. As a result, fine-tuned models can appear far apart under divergences defined over raw probabilities, yielding weak CP-$\Delta$ guarantees, despite differing meaningfully, i.e., in ways that affect the generated token, on only a small set of tokens. We formalize this phenomenon via the Sparse Relative Probability Shift (SpaRPS) property, an idealized abstraction of this structural sparsity. We observe that fine-tuned LLMs may not naturally satisfy this property due to widespread shifts in per-token probabilities induced by fine-tuning. We conjecture that most shifts do not lead to meaningful changes in generated outputs, and thus SpaRPS can be enforced via a smoothing mechanism while preserving downstream performance.

**SCP-$\Delta_r$: enforcing SpaRPS via relative probabilities and smoothing.** Building on this insight, we propose SCP-

$\Delta_r$, a NAF-based algorithm designed to explicitly exploit and enforce SpaRPS for TDE protection.

SCP-$\Delta_r$ combines two complementary components:

1. **Relative probabilities.** Following Segal et al. (2025), we operate on relative probability distributions (RPDs) rather than raw probabilities. This removes global normalization effects and yields dramatically smaller NAF bounds when probability shifts are sparse in relative space, as predicted by SpaRPS.

2. **Base-model smoothing.** To ensure that SpaRPS holds in practice despite widespread probability shifts, we introduce a smoothing mechanism that aligns most token-level relative probabilities with those of a shared base model, preserving only the most informative deviations. This explicitly enforces SpaRPS while incurring minimal utility loss.

Together, these changes enable SCP-$\Delta_r$ to achieve orders-of-magnitude smaller NAF bounds than CP-$\Delta$.

**From NAF guarantees to training data extraction protection.** Crucially, NAF guarantees do not automatically imply protection against training data extraction. We show that CP-$\Delta$ bounds may allow large information gain for a TDE adversary. In contrast, we prove that under reasonable and empirically satisfied assumptions, SCP-$\Delta_r$ yields small bounds on the information gain of a TDE adversary. Attempts to derive comparable guarantees for CP-$\Delta$ result in bounds that do not meaningfully constrain TDE.

**Empirical evaluation.** We evaluate SCP-$\Delta_r$ against CP-$\Delta$ and other NAF-based methods using both existing and newly proposed TDE attacks. Our results show that SCP-$\Delta_r$ achieves the strongest protection against TDE attacks, with no observable utility degradation. In contrast, we find that previously proposed NAF-based methods can be highly vulnerable to TDE attacks.

**Contributions.**

1. We introduce SpaRPS, a property capturing the idealized sparsity structure of fine-tuning-induced relative probability shifts, and show that fine-tuned models can be smoothed to satisfy this property while preserving performance.

2. We propose SCP-$\Delta_r$, a practical NAF-based defense that operates on relative probability distributions and enforces SpaRPS via base-model smoothing, achieving strong theoretical and empirical protection against TDE with minimal utility degradation.

3. We establish the first formal connection between NAF guarantees and training data extraction resistance, showing that existing NAF-based methods may fail to meaningfully constrain TDE, while SCP-$\Delta_r$ yields dramatically smaller extraction bounds under standard assumptions.

4. We demonstrate the first practical extraction attacks against NAF-based methods, and show that previously proposed methods remain practically vulnerable, whereas SCP-$\Delta_r$ achieves substantially stronger empirical and theoretical resistance to these attacks.

In the remainder of the paper, we first discuss relevant background on TDE and NAF. We then introduce the SpaRPS property and empirically study its applicability to LLMs. Next, we present SCP-$\Delta_r$ together with its smoothing mechanism for enforcing SpaRPS, and derive provable guarantees connecting CP-style algorithms to TDE. Finally, we evaluate SCP-$\Delta_r$ against existing NAF-based methods using both adaptive and non-adaptive extraction attacks.

## 2. Background and Related Work

### 2.1. Preliminaries

We use the following notation. Let $T$ denote a training (or fine-tuning) algorithm that maps a dataset $S$ to a conditional generative model $p(\cdot|\cdot)$. For an input $x$ (also referred to as a prompt) and output token $y$, $p(y|x)$ denotes the probability assigned to $y$ given $x$. We occasionally write $p$ to denote a distribution instead of a model. We write $\Delta(p\|q)$ for a divergence between distributions $p$ and $q$, and $D(p, q)$ for a symmetric distance; divergences need not be symmetric. All models and probability distributions are enforced to assign probability of at least $e^{-20}$ to all outputs.

### 2.2. Training Data Extraction

LLMs exhibit a phenomenon known as "eidetic memorization," where they learn to reproduce verbatim sequences from their training corpora to minimize cross-entropy loss on "long-tail" data (Carlini et al., 2021; Tirumala et al., 2022). This creates significant privacy vulnerabilities, as adversaries can extract sensitive information, including Personally Identifiable Information (PII) and proprietary code, by interacting with the model's black-box API (Carlini et al., 2021; Nasr et al., 2025). While safety alignment (e.g., RLHF) was initially thought to mitigate this, recent "divergence attacks" can force models to deviate from their answer style, increasing the rate of training data leakage by orders of magnitude (Nasr et al., 2025).

**Training data extraction of fine-tuned models.** Fine-tuning introduces increased risks as it typically involves

smaller, high-value datasets where the probability shift relative to the base model serves as a strong membership signal. Empirical studies indicate that fine-tuning can amplify PII extraction rates compared to pre-trained baselines (Mireshghallah et al., 2022b; Akkus et al., 2025). Crucially, parameter-efficient methods are not immune; the "LoRA-Leak" framework demonstrates that Low-Rank Adaptation (LoRA) updates still yield high membership inference success (AUC ≈ 0.77) by leveraging the pre-trained base model as a reference (Ran et al., 2025; Marinelli & Eckhoff, 2026).

**Defenses against training data extraction.** Defensive strategies are generally categorized into empirical mitigations and provable guarantees. In-practice methods include training data deduplication and PII scrubbing, though recent work shows that context-based attacks can still reconstruct redacted entities with high accuracy (Lee et al., 2022; Lukas et al., 2023). Training-time interventions like "Goldfish Loss," which randomly masks tokens during loss computation, effectively reduce verbatim memorization but lack formal privacy bounds (Hans et al., 2024). Conversely, Differential Privacy (DP), typically implemented via DP-SGD, provides rigorous mathematical guarantees by clipping gradients and adding noise, but it frequently incurs substantial utility costs and computational overhead that hinder widespread adoption (Dwork, 2006; Abadi et al., 2016; Du et al., 2025). In contrast to all above approaches, we base our method on NAF and achieve both practicality and theoretically grounded protection.

## 2.3. NAF and CP-$\Delta$

NAF constructs a model $p(\cdot|x)$ that remains close, under a divergence $\Delta$, to a set of constituent models $p_1(\cdot|x), \ldots, p_m(\cdot|x)$. The premise is that if, for a given input $x$, at least one constituent model behaves safely, then an aggregate model that is sufficiently close to all constituents should also behave safely. NAF has been applied to copyright protection, access control, poisoning robustness, and privacy (Vyas et al., 2023; Li et al., 2024; Kalai et al., 2025; Segal et al., 2025).

Definitions and theorems of this subsection follow Vyas et al. (2023).

**Definition 2.1** ($k$-NAF). A model $p(\cdot|x)$ is $k_x$-NAF on input $x$ with respect to models $p_1(\cdot|x), \ldots, p_m(\cdot|x)$ and divergence $\Delta$ if, for all $i \leq m$,

$$\Delta\big(p(\cdot|x) \,\|\, p_i(\cdot|x)\big) \leq k_x.$$

If $\Delta\big(p(\cdot|x) \,\|\, p_i(\cdot|x)\big) = k_x$ for all $i$, we call $p(\cdot|x)$ $k_x$-NAF-tight. If $k_x \leq k$ for all $x$, the model is called $k$-NAF.

A common instantiation uses the max divergence $\Delta(p\|q) := \max_y \log \frac{p(y)}{q(y)}$, which implies pointwise probability control (Vyas et al., 2023):

$$p(y|x) \leq e^{k_x} \, p_i(y|x) \quad \text{for all } i, y.$$

**CP-$\Delta$.** A standard NAF method for $m = 2$ is CP-$\Delta$, which takes a pointwise minimum of two distributions and renormalizes:

**Definition 2.2** (CP-$\Delta$). Let $p, q$ be distributions over $Y$. Define

$$\text{CP-}\Delta(p,q)(y) := \frac{\min(p(y), q(y))}{Z},$$

$Z := \sum_{y \in Y} \min(p(y), q(y))$.

**Theorem 2.3** (CP-$\Delta$ bound). *For any input $x$, CP-$\Delta(p(\cdot|x), q(\cdot|x))$ is $k_x$-NAF-tight with respect to $p(\cdot|x)$ and $q(\cdot|x)$ under max divergence, where*

$$k_x = D_m\big(p(\cdot|x), q(\cdot|x)\big), \; D_m(p,q) := \log \frac{1}{1 - \text{TV}(p,q)}$$

*and* $\text{TV}(p,q) := \frac{1}{2} \sum_y |p(y) - q(y)|$.

*Remark* 2.4. The quantity $D_m(p,q)$ can be numerically large: $\text{TV}(p,q)$ may approach 1 even when $p$ and $q$ differ substantially on only a small subset of tokens, yielding guarantees that are not meaningful in practice (Cohen, 2025).

To illustrate how memorization-induced probability spikes affect CP-style bounds, consider the following example:

**Example 2.1.** *Let $p$ and $q$ be probability distributions over $Y = \{0, \ldots, 9\}$ such that $q(y) = 0.1$ for all $y \in Y$, while $p(0) = 0.91$ and $p(y) = 0.01$ for all $y \neq 0$ (See Table 1).*

The distributions in Example 2.1 illustrate a common memorization pattern in fine-tuned LLMs. Here, $p$ concentrates most of its mass on a single memorized token, while $q$ remains balanced. As a result, the pointwise minimum $\min(p(y), q(y))$ is small for most $y$, driving $\text{TV}(p,q)$ close to 1 and yielding a large CP-$\Delta$ bound. This behavior shows how sparse probability spikes, which are characteristic of memorization, can cause CP-style bounds based on raw probabilities to become numerically large.

*Table 1.* Example distributions $p$ and $q$ and their CP-style aggregations. Applying CP to raw probabilities (P) yields a highly imbalanced distribution compared to applying CP to relative probabilities (R). Bold columns indicate the probability representation on which each aggregation operates.

|     | $p$ |     | $q$ |     | CP-$\Delta$ |     | CP-$\Delta_r$ |     |
|     | **P** | R | **P** | R | **P** | R | P | **R** |
| --- | --- | --- | --- | --- | --- | --- | --- | --- |
| 0   | 0.91 | 58 | 0.1 | 1 | 0.53 | 7.9 | 0.15 | 1.5 |
| 1–9 | 0.01 | 0.64 | 0.1 | 1 | 0.05 | 0.79 | 0.09 | 0.96 |

## 2.4. Relative Probabilities and CP-$\Delta_r$

To reduce sensitivity to probability spikes, we follow (Segal et al., 2025) and operate on *relative probability distributions* (RPDs). While (Segal et al., 2025) did not formulate their method within the NAF framework, it can be viewed as applying the CP aggregation rule to relative probabilities rather than to raw probabilities; we refer to this variant as CP-$\Delta_r$. Definitions and theorems of this subsection follow Segal et al. (2025), rephrased in NAF notation.

**Definition 2.5** (RPD). Let $q$ be a distribution over $Y$ and let $n := |Y|$. Define the *typical probability* $tq$ and the corresponding *relative probability distribution* (RPD) $rq$ by

$$tq := \exp\Big(\frac{1}{n}\sum_{y\in Y}\log q(y)\Big),\ rq(y) := \frac{q(y)}{tq}.$$

Conceptually, raw probabilities are sensitive to global normalization effects. Relative probabilities instead isolate changes in the model's relative token preferences, which more directly reflect meaningful behavioral differences between models. Mathematically, RPDs do not sum to 1; instead, they satisfy $\sum_y \log rq(y) = 0$, reducing sensitivity to probability spikes and often yielding smaller bounds in practice (Segal et al., 2025). This construction addresses the failure mode illustrated in Example 2.1, as illustrated in Table 1.

**Definition 2.6** (CP-$\Delta_r$). Let $p, q$ be distributions over $Y$. Define

$$\text{CP-}\Delta_r(p,q)(y) := \frac{\min(rp(y), rq(y))}{Z},$$

where $Z := \sum_{y\in Y}\min(rp(y), rq(y))$.

**Theorem 2.7** (CP-$\Delta_r$ bound). *For any input $x$, $\text{CP-}\Delta_r(p(\cdot|x), q(\cdot|x))$ is $k_x$-NAF-tight with respect to $p(\cdot|x)$ and $q(\cdot|x)$ under the relative max-divergence $\Delta_r(p\|q) := \max_y \log \frac{rp(y)}{rq(y)}$, with*

$$k_x = D_r\big(p(\cdot|x), q(\cdot|x)\big),\ D_r(p,q) := \frac{1}{2n}\sum_y\ \log\frac{rp(y)}{rq(y)}\ .$$

As discussed in Section 4.1, the bound in Theorem 2.7 is dramatically smaller in practice than the bound in Theorem 2.3 (Figure 1b).

## 2.5. Other NAF Variations

CP-style algorithms have also been analyzed under alternative divergences such as KL (Vyas et al., 2023). Recent work observed that sequence generation under CP-$\Delta_{\text{KL}}$ may drift toward one constituent model, compromising safety, and proposed fusing strategies to mitigate this effect (Abad et al., 2025). As discussed in subsection 5.3, such fusing does not address our adaptive extraction attacks.

## 3. The SpaRPS Property

In this section, we introduce the SpaRPS property and discuss its applicability to fine-tuned LLMs. First, we give theoretical justification for using relative probabilities, which SpaRPS operates on.

### 3.1. Why Relative Probabilities

Table 1 illustrates a key failure mode of divergences defined on raw probabilities: even if the difference between two distributions is caused by a single spike, normalization effects can make $p(y)$ and $q(y)$ far apart for all $y$.

Concretely, suppose $p$ and $q$ arise from unnormalized "scores" that differ by an unknown multiplicative constant $C$. A natural way to estimate this constant is to choose $C$ that best aligns $p$ and $q$ in log-space on average. The next theorem shows that, under squared error in log-space, the optimal scaling is exactly the ratio of typical probabilities, which is equivalent to using RPDs.

**Theorem 3.1.** *Let $p$ and $q$ be distributions over $Y$ with $p(y), q(y) > 0$ for all $y \in Y$. Then*

$$\arg\min_{C>0}\ \sum_{y\in Y}\log^2\Big(C\frac{p(y)}{q(y)}\Big) = \frac{tq}{tp}.$$

The proof follows from a standard least-squares argument and is provided in Appendix B.1.

### 3.2. SpaRPS

We are now ready to define the SpaRPS (Sparse Relative Probability Shift) property.

**Definition 3.2** (SpaRPS). Let $p$ and $q$ be distributions over $Y$. We say that $p$ is $m$-*SpaRPS with respect to* $q$ if there exists a set $A \subseteq Y$ with $|A| = m$ and $\alpha > 0$ such that

$$p(y) = \alpha\, q(y) \qquad \forall y \in Y \setminus A.$$

We emphasize that SpaRPS is intended as an idealized structural abstraction rather than a literal property expected to hold exactly for real fine-tuned LLMs without intervention.

**Proposition 3.3.** *Let $n := |Y|$. If $p$ is $m$-SpaRPS with respect to $q$ and $\forall y \in A:\ \log \frac{p(y)}{q(y)}\ \leq \log M$, then*

$$D_r(p,q) \leq \frac{m}{n}\big(\log M + |\log \alpha|\big).$$

*For $m = 20$ and $n = 32000$ (as used in our evaluation), and assuming $\log M + |\log \alpha| \leq 40$ (which follows from all probabilities being lower bounded by $e^{-20}$), we obtain $D_r(p,q) \leq 0.025$. A proof is provided in Appendix B.2.*

By Proposition 3.3 and Theorem 2.7, fine-tuned LLMs that satisfy SpaRPS are close in terms of relative probability

distributions and therefore provide a small CP-$\Delta_r$ bound. However, due to widespread probability shifts induce by fine-tuning, it is unlikely in practice that $p(\cdot|x)$ is exactly $m$-SpaRPS with respect to $q(\cdot|x)$. In particular, the ratio $\frac{p(y|x)}{q(y|x)}$ is unlikely to equal a single constant $\alpha$ for all but $m$ outputs simultaneously. We therefore introduce *Expected SpaRPS* as a more realistic relaxation.

**Definition 3.4** (Expected SpaRPS). Let $T$ be a randomized training algorithm and let $S_p, S_q$ be datasets. For each prompt $x \in X$, define the random conditional models $p(\cdot|x) := T(S_p)(x)$ and $q(\cdot|x) := T(S_q)(x)$, where the randomness is over $T$. Let $n := |Y|$.

We say that $(S_p, S_q)$ satisfies $(m, B, C)$-*Expected SpaRPS on X* if there exists a subset $X' \subseteq X$ with $|X'| \geq C|X|$ such that for every $x \in X'$ there exists a set $A_x \subseteq Y$ with $|A_x| = m$ satisfying

$$|\mathbb{E}[\log rp(y|x)] - \mathbb{E}[\log rq(y|x)]| \leq B \qquad \forall y \in Y \setminus A_x.$$

Expected SpaRPS can be viewed as a PAC-style relaxation of SpaRPS, in which the proportionality condition is allowed to hold approximately (via $B$) and for most prompts (via $C$), under the randomness of the training algorithm. Ideally, $B \approx 0$ and $C \approx 1$.

Figure 1a illustrates the relationship between the Expected SpaRPS parameters $m$, $B$, and $C$ in a practical fine-tuning setting. The datasets $S_p$ and $S_q$ contain 3000 and 2900 training examples respectively, with $S_q \subset S_p$, and the SpaRPS analysis evaluates prompts $X = S_p \setminus S_q$. We estimate $\mathbb{E}[\log rp(y \mid x)]$ and $\mathbb{E}[\log rq(y \mid x)]$ by averaging log relative probabilities over 16 independent fine-tunes (different seeds) of LLaMA2-7B on the *MathAbstracts* dataset (see Section 5.1). We find that even Expected SpaRPS does not hold in practice. As shown in Figure 1a, even when allowing $(m =)1024$ unconstrained shifts, $(1 - C =)10\%$ of prompts still exhibit large deviation ($B \approx 2.1$). This suggests that SpaRPS must be explicitly enforced if it is to serve as the basis for practical TDE guarantees.

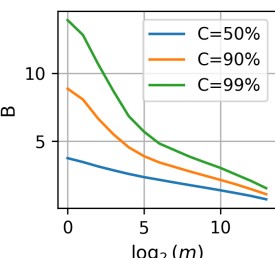

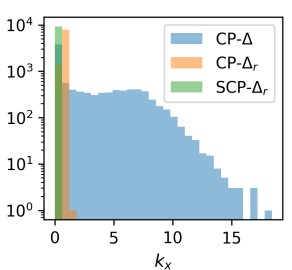

*(a)* Expected SpaRPS analysis  *(b)* $k_x$ histograms

*Figure 1.* Expected SpaRPS and $k_x$ evaluated using the *MathAbstracts* dataset (See 5.1). (a) Expected SpaRPS $B$ as a function of $\log_2(m)$ for different $C$ values. (b) $k_x$ bounds log-scale histograms for different CP style algorithms.

# 4. SCP-$\Delta_r$

We observed that SpaRPS and Expected SpaRPS do not realistically hold in practice. We conjecture, however, that LLMs behave as though they satisfy SpaRPS, in the sense that retaining only a small number of relative probability shifts would not meaningfully affect model behavior. To enforce SpaRPS and enable its use in our theoretical analysis, we introduce the $S$ (smoothing) component of SCP-$\Delta_r$, which enforces the stronger, non-expected form of SpaRPS (Definition 3.2) by replacing most relative probabilities with those of a shared base model $b$ prior to aggregation. Our experiments (Section 5.4) validate our conjecture by showing that the smoothing mechanism of SCP-$\Delta_r$ does not degrade utility.

## 4.1. Base-Model Smoothing

Let $b(\cdot|x)$ denote a fixed *base* model (e.g., the pre-fine-tuning model). Given an RPD $rp$ and a base RPD $rb$, we define smoothing with parameter $m$ via $S_m(rp, rb)$, which preserves the relative probabilities of $m$ selected tokens and sets all remaining tokens to match the base model, up to a global rescaling.

Formally, let $I \subseteq Y$ be a token set of size $m$ (chosen by a scoring rule defined below). Define $S_m(rp, rb)$ by

$$S_m(rp, rb)(y) = \begin{cases} \beta\, rp(y), & y \in I, \\ \beta\, rb(y), & y \notin I, \end{cases}$$

where $\beta > 0$ is chosen s.t. $\sum_{y \in Y} \log S_m(rp, rb)(y) = 0$ (i.e., the result is an RPD). We use $\widetilde{rp}$ to denote a smoothed RPD. Note that for small $m$, $\beta$ is close to 1 (see Appendix B.6).

We choose $I$ as the $m$ tokens with largest score

$$\text{score}_{p,b}(y) := p(y|x) \cdot \log \frac{rp(y|x)}{rb(y|x)},$$

This score is the per-token contribution to the relative KL divergence: $\Delta_{rKL}(p(\cdot|x) \,\|\, b(\cdot|x)) = \sum_{y \in Y} \text{score}_{p,b}(y)$. Selecting the top-$m$ tokens by $\text{score}_{p,b}$ bounds the potential utility degradation caused by smoothing (see Appendix A). *Remark* 4.1. $S_m(rp, rb)$ is $2m$-SpaRPS with respect to $S_m(rq, rb)$ for any RPDs $rp, rq$ and $rb$. A proof is provided in Appendix B.3.

Algorithm 1 describes SCP-$\Delta_r$ with a smoothing parameter $m$. The resulting model is equivalent to CP-$\Delta_r(\widetilde{p}, \widetilde{q})$, where $\widetilde{p}$ and $\widetilde{q}$ are the probability distributions induced by the relative probability distributions $\widetilde{rp}$ and $\widetilde{rq}$. As a result, any theoretical guarantees for CP-$\Delta_r$ apply directly to SCP-$\Delta_r$. Figure 1b shows that the resulting $k_x$ values for SCP-$\Delta_r$ are substantially smaller than those of CP-$\Delta_r$, which in turn are smaller than those of CP-$\Delta$. At the 99$^{\text{th}}$ percentile, the

---

**Algorithm 1** SCP-$\Delta_r$ (for a fixed input)

---

**Input:** Distributions $b, p, q$ and smoothing level $m$.
**Return:** A distribution $r$.

1: $rb \leftarrow \text{RPD}(b); rp \leftarrow \text{RPD}(p); rq \leftarrow \text{RPD}(q)$.
2: $\widetilde{rp} \leftarrow \text{S}_m(rp, rb); \widetilde{rq} \leftarrow \text{S}_m(rq, rb)$.
3: $u(y) \leftarrow \min(\widetilde{rp}(y), \widetilde{rq}(y))$ for all $y \in Y$.
4: **return** $r(y) \leftarrow u(y)/\sum_{y' \in Y} u(y')$.

---

corresponding values are 0.004, 0.98, and 11.6, respectively, reflecting orders-of-magnitude improvements when moving from CP-$\Delta$ to CP-$\Delta_r$, and from CP-$\Delta_r$ to SCP-$\Delta_r$. Note that CP-$\Delta$ bounds $k_x$ under the standard max-divergence on raw probabilities, whereas CP-$\Delta_r$ and SCP-$\Delta_r$ use relative max-divergence; we compare these quantities as they play the same role in the NAF guarantee.

Importantly, achieving these improvements does not incur substantial computational overhead: Training cost scales approximately linearly with the amount of training data, which is partitioned across the two constituent models. During inference, the three forward passes are independent and can therefore be executed in parallel on a single GPU. Moreover, the aggregation operations are lightweight relative to the forward passes themselves, and based on prior empirical evaluations of CP-style algorithms, the resulting wall-clock overhead is expected to be modest in practice (Abad et al., 2025).

## 4.2. SCP-$\Delta_r$ TDE Protection

We now discuss SCP-$\Delta_r$ protection against TDE. Theorems 4.4 and 4.5 below are stated for CP-$\Delta_r$, but apply directly to SCP-$\Delta_r$ by replacing relative probability distributions with their smoothed counterparts, i.e., $rq \rightarrow \widetilde{rq}$.

**TDE framework.** Typically, TDE attacks consist of two stages. First, the adversary queries the target model to obtain candidate training examples, and then applies a membership inference (MI) attack to predict membership for each candidate (Carlini et al., 2021; Nasr et al., 2025). We assume the adversary possesses prefixes of targeted training examples and aims to extract the next token for each prefix. We further assume the adversary has black-box access to the target model's output probabilities and white-box access to a reference model trained in similar manner and on similar data as the target model, which is standard in prior work on MI (Mireshghallah et al., 2022a; Galende et al., 2025).

**Definition 4.2** (Black-box TDE Game). Let $S_e = \{s_1, \ldots, s_n\} \subset S$ and let $r_1$ be an LLM trained on $S$. Let $X_e = \{x_1, \ldots, x_n\}$ be such that, for all $i$, $x_i$ is a prefix of $s_i$. A TDE adversary ALG is given $X_e$, a reference LLM $r_0$, and black-box access to $r_1$, allowing it to query $r_1$ with any $x$ and observe $r_1(y|x)$ for all $y \in Y$. The adversary outputs

a prediction $y_i$ for each $x_i$ or abstains. The success of ALG is measured by its accuracy on predicted tokens as a function of coverage (the fraction of non-abstained predictions), as standard in selective classification (Geifman & El-Yaniv, 2017).

*Remark* 4.3. We restrict the adversary to predicting specified training examples rather than allowing it to generate arbitrary examples. This is because some training examples are inherently easier to extract than others, and an unrestricted adversary would simply focus on the easiest cases. Our formulation instead enables us to reason about the extractability of a given training example.

**Likelihood ratio attack (LiRA).** Without loss of generality, we assume that the adversary first picks a candidate $y_i$ for each $x_i$ (the generation stage) and then decides whether to keep it or abstain (the MI stage). While an optimal strategy for the full TDE game is intractable (Sablay-rolles et al., 2019), it has been shown that under reasonable assumptions (discussed in Appendix C.1), the optimal strategy for the MI stage is thresholding the likelihood ratio $\frac{r_1(y_i|x_i)}{r_0(y_i|x_i)}$ (LiRA) (Sablayrolles et al., 2019; Mireshghallah et al., 2022a; Galende et al., 2025). Although LiRA is derived for the MI stage, the same ratio also characterizes the information gain obtained from observing a candidate token during generation. Consequently, uniformly bounding this ratio limits the maximum advantage attainable by any candidate-generation strategy, and therefore constrains both the generation and MI stages of TDE attacks.

In the simplified case that $r_0 = q$, standard NAF guarantees yield a direct bound on the likelihood ratio. Indeed, for $r_1 = \text{CP-}\Delta(p, q)$, we have $\frac{r_1(y|x)}{q(y|x)} \le e^{k_x}$, as explained in subsection 2.3. For CP-$\Delta_r$, we present the following result:

**Theorem 4.4** (CP-$\Delta_r$ LiRA bound for $r_0 = q$). *Let $r_1 = CP\text{-}\Delta_r(p, q)$, $r_0 = q$ and $k_x$ the bound in Theorem 2.7 then*

$$\frac{r_1(y|x)}{r_0(y|x)} = \frac{r_1(y|x)}{q(y|x)} \le \frac{tr_1(x)}{tq(x)} \cdot e^{k_x} =: e^{t_x},$$

*where $tq(x)$ and $tr_1(x)$ denote the typical probabilities of $q(\cdot|x)$ and $r_1(\cdot|x)$.*

Proof and empirical analysis of Theorem 4.4 are provided in Appendix B.4 and D.3 respectively.

For $r_0 \neq q$, we assume that $r_0 = \text{CP-}\Delta_r(p_0, q)$ for some model $p_0$. We note that the more similar $r_0$ is to $r_1$ (apart from its training set not including the target set $S_e$), the better for the adversary, so this is the "worst case".

**Theorem 4.5** (LiRA bound for $r_0 = \text{CP-}\Delta_r(p_0, q)$). *Let $r_i = CP\text{-}\Delta_r(p_i, q)$ for $i \in \{0, 1\}$, and $k_x$ the corresponding bound from Theorem 2.7 for $r_1$, then*

$$\frac{r_1(y|x)}{r_0(y|x)} \leq \max\left(1, \frac{rq(y|x)}{rp_0(y|x)}\right) \cdot \frac{tr_1(x)}{tr_0(x)} \cdot e^{k_x} =: e^{v_x(y)}$$

A proof for Theorem 4.5 is provided in Appendix B.5.

*Remark* 4.6. While typical probabilities are affected by probability spikes, even if $p_1$ exhibits such spikes, the aggregated model $r_1$ need not, due to the CP aggregation with $q$. As a result, the ratios $\frac{tr_1(x)}{tr_0(x)}$ and $\frac{tr_1(x)}{tq(x)}$ appearing in Theorems 4.4 and 4.5 are expected to be small for most prompts in practice.

*Remark* 4.7 (Theorem 4.5 variation for CP-$\Delta$). Let $r_i = $ CP-$\Delta(p_i, q)$ for $i \in \{0, 1\}$, and $k_x$ the corresponding bound from Theorem 2.3, then $\frac{r_1(y|x)}{r_0(y|x)} \leq \max(1, \frac{q(y|x)}{p_0(y|x)}) \cdot e^{k_x}$. Note that for CP-$\Delta$, $k_x$ is empirically large compared to CP-$\Delta_r$ and SCP-$\Delta_r$ (Figure 1b).

**Proposition 4.8** ($v_x(y)$ is small for smoothed (rare) tokens). *For SCP-$\Delta_r$, suppose that $y$ is smoothed in both $\widetilde{rq}$ and $\widetilde{rp_0}$. Then, under the parameters of Proposition 3.3, $\frac{\widetilde{rq}(y|x)}{\widetilde{rp_0}(y|x)} \leq$ 1.026, which implies*

$$e^{v_x(y)} \leq 1.026 \cdot \frac{tr_1(x)}{tr_0(x)} \cdot e^{k_x} \leq 1.053 \cdot \frac{tr_1(x)}{tr_0(x)}.$$

Notably, under the conditions of Propositions 3.3 and 4.8, together with Remark 4.6, all terms contributing to the bound $e^{v_x(y)}$ are small. A proof and illustration of Proposition 4.8 are provided in Appendix B.6 and D.4, respectively.

Figure 2 illustrates empirical histograms of the bounds implied by Theorem 4.5 in a rare vs typical token scenarios. Here, $p_i = T(S_i)$ and $q = T(S_q)$. The datasets $S_1$, $S_0$, and $S_q$ contain 3000, 2900, and 3000 training examples respectively, with $S_0 \subset S_1$ and $S_1 \cap S_q = \emptyset$. The results in Figure 2 indicate that SCP-$\Delta_r$ substantially improves the extraction bound relative to CP-$\Delta_r$ and CP-$\Delta$, especially for rare tokens, as Proposition 4.8 suggests.

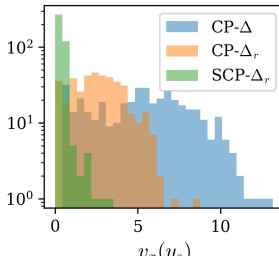

*(a)* Rare tokens extraction

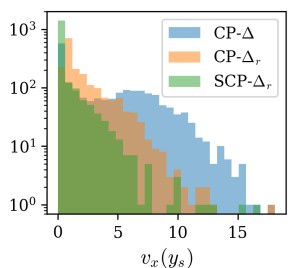

*(b)* Typical tokens extraction

*Figure 2.* Log-scale histograms of Theorem 4.5 bound $v_x(y_s)$, evaluated at the token $y_s$ corresponding to the extraction target $s$. (a) Extraction of rare tokens (Canaries, see 5.3). (b) Extraction of tokens from the *MathAbstracts* dataset (see 5.1).

## 5. Evaluation

### 5.1. Experimental Setup

We evaluate the same datasets as Abad et al. (2025), in addition to a synthesized *Names2IDs* dataset, for a total of four datasets: *MathAbstracts* (Zhang et al., 2025b) (paper titles and abstracts), *StoryTelling* (Fan et al., 2018) (story topics and stories), *CodeInstructions* (Morales, 2023) (instructions and Python solutions), and *Names2IDs* (synthetic names paired with 10-digit IDs). For *CodeInstructions* we fine-tune StarCoder-7B (Li et al., 2023); for the remaining datasets we use LLaMA2-7B (Touvron et al., 2023).

Unless stated otherwise, we fine-tune two constituent models $p$ and $q$ on disjoint partitions of the training set, using 3000 examples per partition (6000 total), for 50 epochs to encourage memorization, following Abad et al. (2025). The smoothing parameter is set to $m = 10$ by default (see Appendix C.3 on this choice). All experiments were conducted on an NVIDIA RTX 6000 GPU with 48 GB of memory. Additional training details are provided in Appendix D.1.

**Compared models.** We compare CP-$\Delta$, CP-$\Delta_{\text{KL}}$ (Vyas et al., 2023), CP-Fuse (Abad et al., 2025), CP-$\Delta_r$ (Segal et al., 2025), and our method SCP-$\Delta_r$. We evaluate two variants of SCP-$\Delta_r$: SCP-$\Delta_r$-b uses the pre-fine-tuning model as the base model, while SCP-$\Delta_r$-c uses a constant logits vector as the base model, obtained by averaging the pre-fine-tuning model's logits over 100 held-out examples. As an additional baseline, we report results for *No CP*, which directly uses the model trained on the first partition (i.e., $p$). For reference, we report the validation-set results of *No CP*, together with their standard error (STE).

### 5.2. Utility Evaluation

Since the fine-tuning datasets of the two constituent models originate from the same source distribution, their output distributions are expected to remain similar under typical usage. Consequently, aggregation is not expected to meaningfully degrade utility. This observation was also validated empirically in prior NAF-based works (Vyas et al., 2023; Abad et al., 2025; Segal et al., 2025). Nevertheless, SCP-$\Delta_r$ introduces an additional smoothing component, motivating a dedicated evaluation of its utility impact. We report next-token accuracy on held-out validation sets for all experiments, and additionally evaluate downstream reasoning and instruction-following performance with and without smoothing (Appendix D.7). We also derive utility degradation bounds for SCP-$\Delta_r$ in Appendix A.

### 5.3. Extraction of NAF-Protected LLMs' Training Data

In this subsection, we empirically evaluate NAF-protected models against TDE attacks. While our theoretical analysis

*Table 2.* Full results table. Best values are in bold. Second best values are underlined.

| | NAS ↓ | Canary ↓ | | PII ↓ | | TTE ↓ | | | | | | Utility ↑ | | |
|---|---|---|---|---|---|---|---|---|---|---|---|---|---|---|
| | All | Math | | IDs | | Math | | Story | | Code | | Math | Story | Code |
| | Mean | Mean | 95% | AEL | FER | AUC | ACC | AUC | ACC | AUC | ACC | ACC | ACC | ACC |
| No CP | .99 | 28 | 31 | 7.4 | .73 | .94 | .84 | .98 | .92 | .96 | .84 | .47 | .37 | .67 |
| CP-$\Delta$ | .47 | 8.8 | 17 | 2.4 | .15 | .90 | .69 | .96 | .80 | .89 | .76 | .50 | .41 | .70 |
| CP-$\Delta_{KL}$ | .76 | 23 | 33 | 4.1 | .34 | .93 | .77 | .97 | .86 | .88 | .80 | **.51** | **.42** | .70 |
| CP-Fuse | .50 | 3.8 | 7.1 | 4.1 | .34 | .93 | .77 | .97 | .86 | .88 | .80 | .48 | .38 | .68 |
| CP-$\Delta_r$ | .22 | 3.6 | 8.0 | 1.9 | .06 | .80 | .50 | .86 | **.58** | .80 | .61 | .50 | .40 | .69 |
| SCP-$\Delta_r$-c | .17 | 1.6 | 3.4 | 1.8 | .05 | .79 | .49 | **.83** | .59 | .81 | .61 | .49 | .40 | .69 |
| SCP-$\Delta_r$-b | **.11** | **1.4** | **3.2** | **.43** | **<.001** | **.76** | **.46** | .84 | .60 | **.78** | **.60** | .50 | .39 | **.72** |
| No CP-val | .002 | 1.4 | 3.6 | .12 | 1e-10 | .66 | .38 | .71 | .45 | .71 | .51 | .47 | .37 | .67 |
| STE | - | .11 | .90 | .01 | - | .01 | .01 | .01 | .01 | .02 | .01 | .01 | .01 | .01 |

considers optimal LiRA attacks, our empirical attacks do not explicitly implement LiRA. Nevertheless, they can be interpreted as instantiating LiRA with a near-ideal reference model, as discussed in Appendix C.2. We note that when the adversary is restricted (e.g., when the reference model is far from ideal), previously proposed MIA attacks have been shown to outperform LiRA (Fu et al., 2023; Zhang et al., 2025a). This occurs because the effectiveness of LiRA degrades under restricted adversaries. In our setting, however, we are unaware of attacks stronger than LiRA, which is optimal under certain assumptions (Appendix C.1).

**The canary method (Carlini et al., 2019).** The canary method consists of inserting sequences of random words to the training set and then evaluating the model memorization of them. We use the *exposure score* defined by Carlini et al. (2019). An exposure score $w$ indicates that the canary ranks among the $2^{-w}$ most probable sequences according to the model. Our canaries consist of 3 randomly generated words. We inserted each canary 3 times into the *MathAbstracts* training set (all in the same partition) and trained the models for 5 epochs. Appendix D.5 further reports results for canaries inserted 1 or 10 times.

**PII extraction.** We use the *Names2IDs* dataset to simulate extraction of personally identifiable information (PII). Given a prompt of the form "Name: <name>, ID:", the adversary queries the model, records the next generated token, appends it to the prompt, and repeats adaptively. We report the average extracted length (AEL) and the full ID extraction rate (FER).

**Token-by-token extraction (TTE).** TTE attempts to reconstruct training examples in two adaptively repeated stages. First, the adversary queries the model using a prefix of a training example and records the next generated token. Second, the adversary predicts whether this token corresponds to the true label by thresholding its probability under the model. We report single token extraction success using the selective classification framework (allowing ab-stention), measuring the accuracy-coverage AUC (standard in selective classification (Geifman & El-Yaniv, 2017)) and the accuracy of the attack when the adversary always accepts the generated token (ACC). We evaluate TTE on the *MathAbstracts*, *StoryTelling*, and *CodeInstruction* datasets.

Importantly, the fusing mechanism of CP-Fuse has no effect on the first generated token (Abad et al., 2025). Since both TTE and our PII extraction method are adaptive and rely only on the first generated token for each prompt, CP-Fuse is effectively reduced to CP-$\Delta_{KL}$ in this setting.

## 5.4. Results

We report the main results and ablation studies below. Additional results, including experiments on LoRA fine-tuned LLMs, are presented in Appendix D.

**Main results.** Table 2 summarizes security under the canary method and PII and TTE extraction attacks, together with next-token accuracy on held-out data. We also report the average normalized attack score (NAS) across the three attacks. Specifically, each attack score (column) is normalized to $[0, 1]$ (excluding the STE row), and NAS is computed by averaging the normalized scores across columns, with TTE scores scaled by a factor of $1/3$ to balance the contribution of all attack types. SCP-$\Delta_r$-b provides the strongest protection under the canary method and PII extraction, and achieves security comparable to CP-$\Delta_r$ on TTE, consistent with Proposition 4.8. SCP-$\Delta_r$-c performs similarly to SCP-$\Delta_r$-b, except on PII extraction, which we analyze further in Appendix C.4. Utility remains comparable across CP-style methods, with only minor differences. Notably, non-RPD-based CP-style algorithms remain highly vulnerable to our attacks, with NAS values ranging from 0.47 to 0.76. In contrast, SCP-$\Delta_r$-b achieves an NAS of 0.11, substantially closer to the reference baseline (No CP-val, NAS = 0.002) and substantially improving over CP-$\Delta_r$ (NAS = 0.22).

**Effect of smoothing level.** Figure 3a shows the effect of the smoothing parameter $m$ on the canary score for SCP-$\Delta_r$.

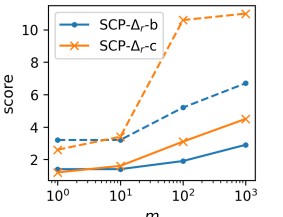
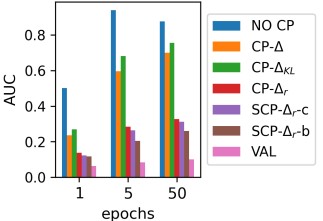

*(a)* Smoothing level effect      *(b)* Memorization effect

*Figure 3.* (a) Canary score vs (log scale) smoothing parameter $m$ (mean: solid, 99[th] percentile: dashed). (b) TTE AUC across different memorization levels on the *MathAbstracts* dataset.

As we increase $m$, the canary tokens are less likely to be smoothed, and thus protection decreases, which supports Proposition 4.8 (See Appendix C.4 for further discussion). In terms of utility, the only noticeable degradation occurs for $m = 1$, for which both variations achieves a validation accuracy of 0.39. For $m \in \{10, 100, 1000\}$, validation accuracy remains stable at $0.495 \pm 0.005$.

**Effect of constituent models memorization.** Figure 3b shows the effect of the number of training epochs on the TTE attack for the *MathAbstracts* dataset. We focus on tokens that $q$ fails to predict, since tokens that $q$ predicts are trivial to recover and hinder distinguishability between methods for low memorization. Notably, the ranking of the different algorithms by AUC is consistent across memorization levels.

**Combining smoothing with raw probabilities.** Applying CP-$\Delta$ to $\widetilde{p}, \widetilde{q}$ being raw probability distributions corresponding to the smoothed RPDs $\widetilde{rp}, \widetilde{rq}$ yields a canary score mean and 95[th] percentile of 8.1 and 18.3, respectively. These values are comparable to those obtained by CP-$\Delta$ (Table 2), indicating that aggregation over RPDs is required for smoothing to achieve a meaningful reduction in extraction.

## 6. Discussion

**Relevance to Pre-Training.** The principles underlying SCP-$\Delta_r$ may be relevant to pre-training. At the same time, pre-training differs in scale, optimization dynamics and data heterogeneity. We therefore view SCP-$\Delta_r$ as a fine-tuning defense, while its extension to pre-training remains an interesting direction for future work.

**Assuming Partition Structure.** Our guarantees rely on the standard partition assumption used by NAF-based methods: for any sensitive example, at least one constituent model is not trained on that example or its semantic equivalent. This assumption is common in prior work on safe aggregation and copyright protection (Vyas et al., 2023; Abad et al., 2025). Moreover, Abad et al. (2025) showed

that CP-style algorithms can empirically retain a comparable level of protection even when the dataset partitions largely overlap (e.g., up to 33%). Extending NAF-based methods to settings with near-duplicates in the training data remains an important direction for future work; one potential approach is to group semantically similar examples within the same partition.

**Reliance on Likelihood-Ratio Optimality Assumptions.** Our extraction bounds do not assume a specific adversarial strategy. Instead, they provide uniform bounds on the likelihood ratio $\frac{r_1(y|x)}{r_0(y|x)}$. This likelihood ratio is the optimal test statistic for membership inference under reasonable assumptions, and also captures the information gain relevant to candidate generation. While the bounds themselves hold independently of how the adversary generates or selects candidates, their interpretation as effective constraints on TDE advantage relies on this connection to optimality. It remains an open question in which regimes these assumptions hold, and whether attacks meaningfully stronger than LiRA exist when they do not.

## 7. Conclusion

Fine-tuning can amplify memorization and enable training data extraction (TDE) attacks, yet existing defenses either lack formal guarantees or incur substantial utility loss. In particular, CP-style NAF guarantees are often insufficient in realistic fine-tuning settings due to probability spikes.

To address this, we introduced the SpaRPS property to capture sparsity in relative probability shifts and proposed SCP-$\Delta_r$, a NAF-based algorithm that operates on relative probability distributions and enforces SpaRPS via base-model smoothing. Our method achieves orders-of-magnitude stronger TDE guarantees than existing CP-style methods and empirically reduces canary exposure and PII extraction success while preserving utility.

Overall, our results demonstrate that our combination of relative probabilities and targeted smoothing makes NAF-style defenses practically effective for protecting fine-tuned LLMs against training data extraction. In contrast, previously proposed NAF-style defenses are practically vulnerable to our attacks. Extending this approach to relaxed assumptions and broader training regimes remains an important direction for future work.

## Impact Statement

This paper presents methods for protecting fine-tuned large language models from training data extraction attacks by providing strong theoretical and empirical privacy guarantees. The primary intended impact of this work is to improve the safety of machine learning systems trained on

proprietary or sensitive data, including personal information, confidential documents, and private codebases.

By reducing the risk of unintended memorization and extraction, our results may help enable more responsible deployment of fine-tuned language models in settings where privacy concerns currently limit their use. Overall, this work advances the technical foundations of privacy-preserving machine learning, and its broader societal implications are largely positive and well understood.

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

## A. Bounding SCP-$\Delta_r$ Utility Degradation

For $r = \text{CP-}\Delta(p, q)$, Vyas et al. (2023) proved that $TV(r, q) \leq TV(q, p)$ (and by symmetry, $TV(r, p) \leq TV(q, p)$). This implies that if $p$ and $q$ are similar in total variation, then $r$ is also close to both, and therefore its utility should be comparable.

For SCP-$\Delta_r$, utility degradation may arise from two sources: (i) the smoothing mechanism, and (ii) the aggregation of relative probability distributions (RPDs). In the following, we derive separate bounds for each source of potential degradation, beginning with the aggregation stage.

**Lemma A.1** (Utility degradation bound - aggregation). *Let $r = \text{CP-}\Delta_r(p, q)$. Then*

$$TV(r, q) \leq TV(p, q).$$

Proof is provided in Appendix B.7.

**Lemma A.2** (Utility degradation bound - smoothing). *Let $\widetilde{p}$ be the probability distributions corresponding to the RPD $\widetilde{rp} = S_m(rp, rb)$ (see Algorithm 1) and $A$ be a set of $m$ "unsmoothed" tokens and $\bar{A}$ its complement. Let $\epsilon = \frac{tp}{tb} \sum_{y \in \bar{A}} b(y)$. Then*

$$\Delta_{\text{KL}}(p\|\widetilde{p}) \leq \log(1 + \epsilon) + \sum_{y \in \bar{A}} p(y) \log \frac{rp(y)}{rb(y)}.$$

*Consequently, by Pinsker's inequality, $TV(p, \widetilde{p}) \leq \sqrt{\frac{1}{2}\Delta_{\text{KL}}(p\|\widetilde{p})}$.*

Proof is provided in Appendix B.8.

*Remark* A.3. The smoothing operator selects the set $A$ as

$$A \in \arg \max_{|A|=m} \sum_{y \in A} p(y) \log \frac{rp(y)}{rb(y)}.$$

Equivalently, since $\sum_y p(y) \log \frac{rp(y)}{rb(y)}$ is constant with respect to $A$, this is the same as

$$A \in \arg \min_{|A|=m} \sum_{y \in \bar{A}} p(y) \log \frac{rp(y)}{rb(y)}.$$

Thus, $A$ is the optimal choice of size $m$ for minimizing the upper bound on $\Delta_{\text{KL}}(p\|\widetilde{p})$ given in Lemma A.2.

*Remark* A.4. Since $p$ is a fine-tuned model and $b$ is a base model, we typically expect $tp \lesssim tb$ (more probability spikes implies lower typical probability). There are two common regimes in which $\epsilon \ll 1$.

First, if $\sum_{y \in \bar{A}} b(y) \ll 1$, which occurs when $b$ is close to $p$ and the unsmoothed $m$ tokens are sufficient to capture most of the probability mass.

Second, if $tp \ll tb$, which occurs when $p$ exhibits probability spikes that concentrate mass on a small number of tokens, thereby reducing the mass assigned to typical tokens.

## B. Proofs

### B.1. Proof of Theorem 3.1

*Proof.* Let $a_y = \log \frac{p(y)}{q(y)}$. Then

$$\sum_y \log^2 \left( C \frac{p(y)}{q(y)} \right) = \sum_y (\log C + a_y)^2.$$

It is well known that the minimizer of $\sum_i (x + a_i)^2$ over $x$ is $x = -\frac{1}{n} \sum_i a_i$. Applying this yields

$$\log C = -\frac{1}{n} \sum_y \log \frac{p(y)}{q(y)} = \log \frac{tq}{tp},$$

and therefore $C = \frac{tq}{tp}$. $\qquad\square$

## B.2. Proof of Proposition 3.3

*Proof.* Let $A \subseteq Y$ satisfy $|A| = m$ and $p(y) = \alpha q(y)$ for all $y \in Y \setminus A$, with $\alpha > 0$. For $y \notin A$ we have $\log q(y) - \log p(y) = -\log \alpha$, and therefore

$$\log \frac{\alpha tq}{tp} = \log \alpha + \frac{1}{n} \sum_{y \in Y} \big( \log q(y) - \log p(y) \big) = \frac{m}{n} \log \alpha + \frac{1}{n} \sum_{y \in A} \big( \log q(y) - \log p(y) \big).$$

Hence, using $\log \frac{p(y)}{q(y)} \leq \log M$ for all $y \in A$,

$$\log \frac{\alpha tq}{tp} \leq \frac{m}{n} |\log \alpha| + \frac{1}{n} \sum_{y \in A} \log \frac{q(y)}{p(y)} \leq \frac{m}{n} \big( |\log \alpha| + \log M \big).$$

Next,

$$D_r(p, q) = \frac{1}{2n} \sum_{y \in Y} \log \frac{p(y)}{\alpha q(y)} + \log \frac{\alpha tq}{tp} \leq \frac{1}{2n} \sum_{y \in Y} \log \frac{p(y)}{\alpha q(y)} + \frac{1}{2} \log \frac{\alpha tq}{tp}.$$

For $y \notin A$, $\log \frac{p(y)}{\alpha q(y)} = 0$, and for $y \in A$,

$$\log \frac{p(y)}{\alpha q(y)} = \log \frac{p(y)}{q(y)} - \log \alpha \leq \log \frac{p(y)}{q(y)} + |\log \alpha| \leq \log M + |\log \alpha|.$$

Thus,

$$\sum_{y \in Y} \log \frac{p(y)}{\alpha q(y)} = \sum_{y \in A} \log \frac{p(y)}{\alpha q(y)} \leq m \big( \log M + |\log \alpha| \big).$$

Combining the bounds yields

$$D_r(p, q) \leq \frac{m}{2n} \big( \log M + |\log \alpha| \big) + \frac{1}{2} \cdot \frac{m}{n} \big( \log M + |\log \alpha| \big) = \frac{m}{n} \big( \log M + |\log \alpha| \big).$$

□

## B.3. Proof of Remark 4.1

*Proof.* Let $sp = \mathrm{S}_m(rp, rb)$ and $sq = \mathrm{S}_m(rq, rb)$. Let $A_p \subseteq Y$ denote the set of indices whose outputs in $sp$ originate from $rp$, and define $A_q \subseteq Y$ analogously for $sq$. By construction of the smoothing operator, $|A_p| = |A_q| = m$.

For all $y \in Y \setminus A_p$, there exists a constant $\alpha_p > 0$ such that $sp(y) = \alpha_p \, rb(y)$. Similarly, for all $y \in Y \setminus A_q$, there exists a constant $\alpha_q > 0$ such that $sq(y) = \alpha_q \, rb(y)$. Therefore, for all $y \in Y \setminus (A_p \cup A_q)$,

$$sp(y) = \alpha_p \, rb(y) = \frac{\alpha_p}{\alpha_q} \, sq(y).$$

Let $A := A_p \cup A_q$. Since $|A| \leq 2m$, if $|A| < 2m$ we enlarge $A$ arbitrarily to a superset $A' \supseteq A$ with $|A'| = 2m$. The proportionality above continues to hold for all $y \in Y \setminus A'$. Hence $sp$ is $2m$-SpaRPS with respect to $sq$. □

## B.4. Proof of Theorem 4.4

.

*Proof.* Let $rr_1 := \mathrm{RPD}(r_1)$ and $rq := \mathrm{RPD}(q)$, i.e., $rr_1(y|x) = r_1(y|x)/tr_1(x)$ and $rq(y|x) = q(y|x)/tq(x)$. Then for any $y$,

$$\frac{r_1(y|x)}{q(y|x)} = \frac{rr_1(y|x)}{rq(y|x)} \cdot \frac{tr_1(x)}{tq(x)} \leq e^{k_x} \cdot \frac{tr_1(x)}{tq(x)}.$$

□

## B.5. Proof of Theorem 4.5

*Proof.* By definition of CP-$\Delta_r$, for $i \in \{0, 1\}$,

$$r_i(y) = \frac{\min(rp_i(y), rq(y))}{Z_i}, \qquad Z_i = \sum_y \min(rp_i(y), rq(y)).$$

Hence,

$$\frac{r_1(y)}{r_0(y)} = \frac{\min(rp_1(y), rq(y))}{\min(rp_0(y), rq(y))} \cdot \frac{Z_0}{Z_1}.$$

Since $\min(rp_1(y), rq(y)) \leq rq(y)$,

$$\frac{r_1(y)}{r_0(y)} \leq \frac{rq(y)}{\min(rp_0(y), rq(y))} \cdot \frac{Z_0}{Z_1} = \max\left(1, \frac{rq(y)}{rp_0(y)}\right) \cdot \frac{Z_0}{Z_1}.$$

Let $rr_i := \mathrm{RPD}(r_i) = r_i / tr_i$. Then

$$rr_i(y) = \frac{r_i(y)}{tr_i} = \frac{\min(rp_i(y), rq(y))}{Z_i \, tr_i},$$

so $\frac{rr_i(y)}{\min(rp_i(y), rq(y))} = \frac{1}{Z_i tr_i}$ is constant in $y$. By the tight form of Theorem 2.7 (i.e., the maximum is attained as an equality),

$$\max_y \frac{rr_i(y)}{\min(rp_i(y), rq(y))} = e^{D_r(p_i, q)},$$

and therefore (since the ratio is constant in $y$)

$$\frac{1}{Z_i tr_i} = e^{D_r(p_i, q)} \qquad \Longrightarrow \qquad Z_i = \frac{e^{-D_r(p_i, q)}}{tr_i}.$$

Thus,

$$\frac{Z_0}{Z_1} = \frac{tr_1}{tr_0} \cdot e^{D_r(p_1, q) - D_r(p_0, q)} \leq \frac{tr_1}{tr_0} \cdot e^{k_x},$$

where $k_x := D_r(p_1, q)$ and $D_r(p_0, q) \geq 0$. Substituting back completes the proof. $\qquad\square$

## B.6. Proof of Proposition 4.8

*Proof.* Let $b$ be the base model and let $\widetilde{rq} = \mathrm{S}_m(rq, rb)$ and $\widetilde{rp_0} = \mathrm{S}_m(rp_0, rb)$ denote the smoothed RPDs. By assumption, $y$ is smoothed in both, hence

$$\widetilde{rq}(y|x) = \beta_q(x)\, rb(y|x), \qquad \widetilde{rp_0}(y|x) = \beta_0(x)\, rb(y|x),$$

for some normalization constants $\beta_q(x), \beta_0(x) > 0$ chosen so that the resulting distributions are valid RPDs (i.e., have zero mean log).

Let $A_q(x)$ (resp. $A_0(x)$) be the unsmoothed token set of size $m$ for $q$ (resp. $p_0$) at input $x$. From the RPD normalization constraint $\sum_{z \in Y} \log \widetilde{rq}(z|x) = 0$ and $\sum_{z \in Y} \log rb(z|x) = 0$, we obtain

$$n \log \beta_q(x) = \sum_{z \in A_q(x)} \Big( \log rb(z|x) - \log rq(z|x) \Big),$$

and similarly,

$$n \log \beta_0(x) = \sum_{z \in A_0(x)} \Big( \log rb(z|x) - \log rp_0(z|x) \Big),$$

where $n := |Y|$.

Assume a bounded log-ratio condition on the unsmoothed tokens, namely that for all relevant $x$ and all $z \in A_q(x)$, $|\log rb(z|x) - \log rq(z|x)| \le B$, and analogously for $A_0(x)$. Then

$$|\log \beta_q(x)| \le \frac{mB}{n}, \qquad |\log \beta_0(x)| \le \frac{mB}{n}.$$

Therefore,

$$\frac{\widetilde{rq}(y|x)}{\widetilde{rp_0}(y|x)} = \frac{\beta_q(x)\, rb(y|x)}{\beta_0(x)\, rb(y|x)} = \exp\Big( \log \beta_q(x) - \log \beta_0(x) \Big) \le \exp\Big( |\log \beta_q(x)| + |\log \beta_0(x)| \Big) \le \exp\Big( \tfrac{2mB}{n} \Big).$$

Note that since all probabilities are lower bounded by $e^{-20}$, all typical probabilities are also lower bounded by $e^{-20}$. Consequently, all relative probabilities lie in the range $[e^{-20}, , e^{20}]$, implying $B = 40$. Plugging in $m = 10$ and $n = 32000$ gives $\exp(2mB/n) = \exp(800/32000) < 1.026$, completing the proof. $\qquad\square$

### B.7. Proof of Lemma A.1

*Proof.* For distributions $u, v$, $TV(u, v) = 1 - \sum_y \min(u(y), v(y))$. Write

$$r(y) = \frac{\min\left(\frac{p(y)}{tp}, \frac{q(y)}{tq}\right)}{z} = \min\left(\frac{p(y)}{ztp}, \frac{q(y)}{ztq}\right), \quad z = \sum_y \min\left(\frac{p(y)}{tp}, \frac{q(y)}{tq}\right).$$

Note that

$$ztp = \sum_y \min\left(p(y), q(y)\frac{tp}{tq}\right) \le \sum_y p(y) = 1, \qquad ztq = \sum_y \min\left(p(y)\frac{tq}{tp}, q(y)\right) \le \sum_y q(y) = 1.$$

Hence $\frac{p(y)}{ztp} \ge p(y)$ and $\frac{q(y)}{ztq} \ge q(y)$ for all $y$. Let $A_y = \frac{p(y)}{ztp}$ and $B_y = \frac{q(y)}{ztq}$. Then $r(y) = \min(A_y, B_y)$ and $B_y \ge q(y)$, so

$$\min(r(y), q(y)) = \min(A_y, B_y, q(y)) = \min(A_y, q(y)) \ge \min(p(y), q(y)).$$

Summing over $y$ yields $\sum_y \min(r(y), q(y)) \ge \sum_y \min(p(y), q(y))$, hence $TV(r, q) \le TV(p, q)$. $\qquad\square$

### B.8. Proof of Lemma A.2

*Proof.* Let $rp(y) = p(y)/tp$ and $rb(y) = b(y)/tb$ with $tp, tb > 0$. Let $A$ be a set of $m$ "unsmoothed" tokens and $\bar{A}$ its complement. Define the smoothed RPD

$$\widetilde{rp}(y) = \beta \begin{cases} rp(y) & y \in A, \\ rb(y) & y \in \bar{A}, \end{cases}$$

where $\beta > 0$ normalizes $\widetilde{rp}$ into an RPD. After converting back to probabilities: $\widetilde{p}(y) = \widetilde{rp}(y)\, \widetilde{tp}$ for some $\widetilde{tp} > 0$ (so that $\sum_y \widetilde{p}(y) = 1$). Let $\epsilon = \frac{tp}{tb} \sum_{y \in \bar{A}} b(y)$. Since $\sum_y \widetilde{p}(y) = 1$ and $\widetilde{p}(y) = \beta \widetilde{tp}\, rp(y)$ for $y \in A$ and $\widetilde{p}(y) = \beta \widetilde{tp}\, rb(y)$ for $y \in \bar{A}$, we have

$$1 = \beta \widetilde{tp}\Big( \sum_{y \in A} rp(y) + \sum_{y \in \bar{A}} rb(y) \Big) \quad \Rightarrow \quad \frac{1}{\beta \widetilde{tp}} = \sum_{y \in A} rp(y) + \sum_{y \in \bar{A}} rb(y).$$

Rewrite the RHS using $rp = p/tp$ and $rb = b/tb$:

$$\frac{1}{\beta \widetilde{tp}} = \frac{1}{tp} \sum_{y \in A} p(y) + \frac{1}{tb} \sum_{y \in \bar{A}} b(y) \le \frac{1}{tp}\Big( 1 + \frac{tp}{tb} \sum_{y \in \bar{A}} b(y) \Big) = \frac{1 + \epsilon}{tp}.$$

Hence $\beta \widetilde{tp} \ge \frac{tp}{1+\epsilon}$.

If $y \in A$ then

$$\widetilde{p}(y) = \beta \widetilde{tp}\, rp(y) \ge \frac{tp}{1+\epsilon} \cdot \frac{p(y)}{tp} = \frac{p(y)}{1+\epsilon},$$

so $\log \frac{p(y)}{\widetilde{p}(y)} \leq \log(1 + \epsilon)$.

If $y \in \bar{A}$ then

$$\widetilde{p}(y) = \beta \widetilde{tp} \, rb(y) \geq \frac{tp}{1 + \epsilon} \cdot \frac{b(y)}{tb},$$

so

$$\frac{p(y)}{\widetilde{p}(y)} \leq (1 + \epsilon) \frac{p(y)tb}{b(y)tp} = (1 + \epsilon) \frac{rp(y)}{rb(y)},$$

and thus $\log \frac{p(y)}{\widetilde{p}(y)} \leq \log(1 + \epsilon) + \log \frac{rp(y)}{rb(y)}$.

Summing over $A$ and $\bar{A}$ yields

$$\Delta_{\mathrm{KL}}(p\|\widetilde{p}) \leq \log(1 + \epsilon) + \sum_{y \in \bar{A}} p(y) \log \frac{rp(y)}{rb(y)}.$$

Pinsker gives the stated TV bound. $\qquad\square$

## C. Further discussions

### C.1. A TDE Optimal Strategy

The likelihood-ratio attack (LiRA) has been shown to be optimal for membership inference (MI) under a set of standard assumptions. We state these assumptions informally below; for precise formulations and proofs, see (Sablayrolles et al., 2019):

1. The distribution over model parameters $\theta$ is assumed to follow a fixed likelihood model determined by the training loss induced by $\theta$.

2. The reference model $r_0$ is trained using the same procedure and on data drawn from a distribution similar to that of $r_1$.

3. Candidate members $(x_i, y_i)$ are drawn i.i.d. from the data distribution, each having probability $\lambda$ of being a member of the training set.

While Assumption 3 does not generally hold for all TDE attacks, LiRA can nevertheless remain relevant even when this assumption is violated; see Appendix C.2.

### C.2. Relevance of LiRA to Our Attacks

Our attacks do not explicitly rely on a reference model and are therefore potentially weaker than the optimal likelihood-ratio attack (LiRA) analyzed in our theory. Nevertheless, LiRA remains closely connected to our attack mechanisms. Below, we clarify this connection for each attack setting.

**Canary method.** Our canaries consist of sequences of three random words. An ideal reference model would assign nearly uniform probability across such sequences, substantially limiting the usefulness of an explicit reference model. In this case, LiRA effectively reduces to thresholding the target model's output probabilities, making reference-model-free attacks comparable in strength.

**PII extraction.** For PII extraction, an ideal reference model would assign uniform probability to all digits. In this case $r_0(y)$ is constant over candidate digits, so the likelihood ratio $\frac{r_1(y|x)}{r_0(y|x)}$ differs from $r_1(y|x)$ only by a constant scaling. Thus, using LiRA is equivalent to thresholding $r_1(y|x)$ after adjusting the decision threshold.

**Token-by-token extraction (TTE).** In TTE, candidate tokens are not drawn from a fixed pool but arise naturally from the language model's generation process. As a result, some of the assumptions under which LiRA is optimal for the membership inference stage do not strictly hold, in particular Assumption 3 in Appendix C.1. For example, given the prefix "12345," the next token "6" is highly likely due to language structure alone, even if the likelihood ratio between the target and reference models is small. In such cases, thresholding the target model probability can be more appropriate than directly applying

LiRA. Nevertheless, as discussed in Section 4.2, the likelihood ratio still quantifies the information gain an adversary obtains from observing the target model's output. Bounding this ratio therefore constrains the adversary's advantage in the candidate generation stage, which makes LiRA-based analysis relevant for TTE as well.

### C.3. Choice of Smoothing Parameter $m$

We did not perform hyperparameter tuning to select the smoothing parameter $m$. Instead, we fix $m = 10$ throughout all experiments except in the ablation study that examines the effect of varying $m$. This choice reflects a trade-off between expressiveness and robustness: on the one hand, $m$ is small relative to the vocabulary size $n$, which is typically on the order of tens of thousands of tokens; on the other hand, retaining ten tokens appears sufficient to preserve the dominant contributors to the model utility in practice, as validated by our utility metric.

While performance may be further improved by tuning $m$ for a specific dataset or training regime, our results suggests that $m = 10$ serves as a reasonable default that does not rely on dataset-specific or training-specific details.

### C.4. Theoretical Analysis of SCP-$\Delta_r$-b vs. SCP-$\Delta_r$-c

As shown in Table 2, SCP-$\Delta_r$-b and SCP-$\Delta_r$-c achieve comparable performance on the canary method and TTE, while SCP-$\Delta_r$-b provides stronger protection on PII extraction. This difference can be explained by Proposition 4.8.

By the proposition, SCP-$\Delta_r$ achieves its strongest advantage over CP-$\Delta_r$ on tokens that are smoothed in both $\widetilde{rq}$ and $\widetilde{rp_0}$. In the PII setting, the tokens of interest are the digits 0-9. Recall that the smoothing operator retains the $m$ tokens with the highest relative information gain relative to the base model and smooths the remaining tokens.

For SCP-$\Delta_r$-b, the base model is the pre-fine-tuning LLM, which is context-aware. As a result, for digit tokens $d$, it is common that $rq(d) \approx rb(d)$, leading to low relative information gain. Consequently, digit tokens are likely to be smoothed, yielding stronger protection.

In contrast, SCP-$\Delta_r$-c uses a non-contextual base model that assigns a constant distribution. Such a base model assigns relatively low probability to digit tokens compared to a context-aware distribution, resulting in high relative information gain for digits. These tokens are therefore less likely to be smoothed, weakening protection on PII extraction.

Finally, note that this reasoning applies when the number of relevant tokens is on the order of or smaller than $m$. In the canary setting, however, the number of candidate tokens is much larger than $m$, making the probability that any specific target token is smoothed high regardless of whether the base model is contextual. As illustrated in Figure 3a, increasing $m$ makes this effect apparent for the canary method as well.

## D. Additional Training parameters and Experiments

### D.1. Training Details

We follow the training setup of Abad et al. (2025). Our default training hyperparameters are reported in Table 3. We use a batch size of 1, since training examples are concatenated with an EOS (end-of-sequence) token up to a maximum sequence length of 2048.

The fine-tuning process requires adding a separator token and a padding token to the tokenizer. We found that the default Hugging Face initialization for newly added tokens led to gradient explosions in our setting. Instead, we initialize the embeddings of new tokens using the EOS token embedding, which we found to be stable.

The LoRA training parameters are reported in Table 3. For LoRA fine-tuning, concatenating training examples dramatically reduced memorization in the constituent models, preventing meaningful evaluation. We therefore use a sequence length of 25, corresponding to the maximum length of examples in the *Names2IDs* dataset (shorter examples are padded with EOS). Additionally, we observed that adding new tokens interacts poorly with LoRA fine-tuning; as a result, we removed the separator token from the dataset and did not introduce new tokens into the tokenizer. Finally, training for 50 epochs led to gradient instability in some runs, so we reduced the number of epochs to 20, which still resulted in substantial memorization.

*Table 3.* Default training parameters for regular fine-tuning and LoRA.

| Parameter | Regular Fine-Tuning Default | LoRA Default |
|---|---|---|
| #train sample per partition | 3000 | 3000 |
| epochs | 50 | 20 |
| batch_size | 1 | 64 |
| neftune_noise_alpha | 5 | None |
| learning_rate | 5e-5 | 5e-4 |
| gradient_accumulation_steps | 1 | 1 |
| optim | adamw_bnb_8bit | adamw_bnb_8bit |
| warmup_steps | 50 | 50 |
| max_grad_norm | 2.0 | 2.0 |
| lora_r | - | 64 |
| lora_alpha | - | 32 |
| lora_dropout | - | 0.05 |
| target_modules | - | q_proj, k_proj, v_proj, o_proj, gate_proj, up_proj, down_proj, lm_head |

*Table 4.* 99th percentile $k_x$ values for the *MathAbstracts* (M), *StoryTelling* (S) and *CodeInstructions* (C) datasets for different CP-style algorithms.

| | $k_x$ (99%) ↓ | | |
|---|---|---|---|
| | M | S | C |
| CP-$\Delta$ | 11.6 | 12.6 | 14.0 |
| CP-$\Delta_r$ | .98 | 1.58 | .84 |
| SCP-$\Delta_r$-b | **.004** | **.005** | **.003** |
| SCP-$\Delta_r$-c | .005 | .008 | .004 |

## D.2. $k_x$ Bounds

Table 4 reports the 99th percentile of the $k_x$ bound on the *MathAbstracts*, *StoryTelling*, and *CodeInstructions* datasets. SCP-$\Delta_r$ achieves substantially smaller $k_x$ values than CP-$\Delta_r$, which in turn outperforms CP-$\Delta$. The difference between SCP-$\Delta_r$-b and SCP-$\Delta_r$-c is minor, with a slight advantage for SCP-$\Delta_r$-b.

## D.3. $t_x$ Bounds

Figure 4 presents empirical histograms of the $t_x$ bounds (Theorem 4.4) for different CP-style algorithms. SCP-$\Delta_r$ achieves the smallest $t_x$ bounds, followed by CP-$\Delta_r$.

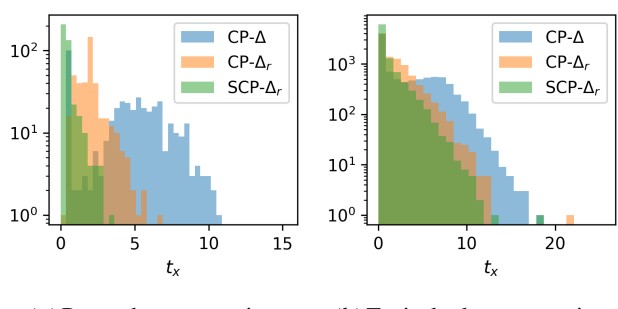

*(a)* Rare tokens extraction    *(b)* Typical tokens extraction

*Figure 4.* Log-scale histograms of Theorem 4.4 bound $t_x$. (a) Extraction of rare tokens (Canaries, see 5.3). (b) Extraction of tokens from the *MathAbstracts* dataset (see 5.1).

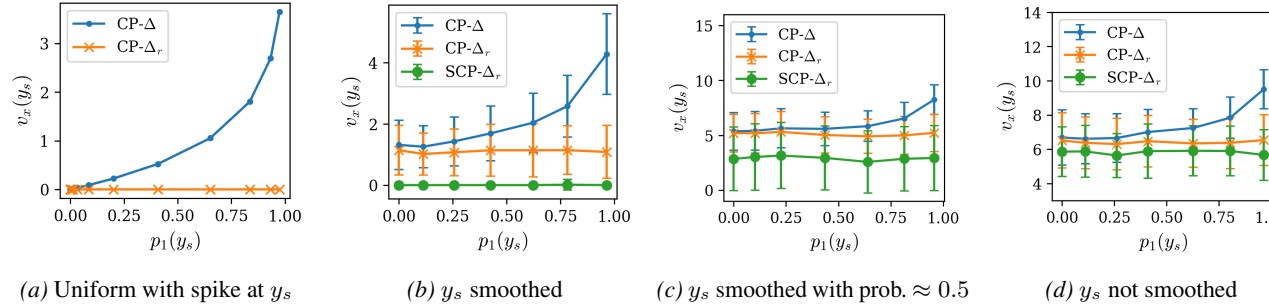

*(a) Uniform with spike at $y_s$*     *(b) $y_s$ smoothed*     *(c) $y_s$ smoothed with prob. $\approx 0.5$*     *(d) $y_s$ not smoothed*

*Figure 5.* Theorem 4.5 bound $v_x(y_s)$ as a function of $p_1(y_s)$ under different smoothing conditions. Error bars denote one standard deviation. (a) $q$ and $p_0$ are uniform, while $p_1$ is uniform except for a spike at $y_s$. (b) $q$, $p_0$, $p_1$ and $b$ are randomly initialized, with $p_1$ additionally spiked at $y_s$. (c) Same as (b), but $q(y_s)$ is scaled so that $y_s$ is smoothed with probability approximately 0.5. (d) Same as (b), but $q(y_s) = \max_y q(y)$, making $y_s$ very unlikely to be smoothed.

### D.4. Illustration of Proposition 4.8

Proposition 4.8 implies that the bound $v_x(y)$ from Theorem 4.5 improves when the target token $y$ is smoothed. To illustrate this effect, we conduct the following synthetic experiment.

We fix a target token $y_s$ and initialize distributions $p_0$, $p_1$, $q$, and the base model distribution $b$. We then gradually increase $p_1(y_s)$ and measure the resulting bound $v_x(y_s)$ for CP-$\Delta$, CP-$\Delta_r$, and SCP-$\Delta_r$. Different initialization schemes are used to control the likelihood that $y_s$ is smoothed. We also include a simple baseline where all distributions are initialized to be uniform (Figure 5a), in which case CP-$\Delta_r$ and SCP-$\Delta_r$ coincide.

For the non-uniform settings, we first sample a shared logits vector $w \sim \mathcal{N}(0, 1)$ to induce correlation between models, and then initialize each distribution $p \in \{p_0, p_1, q, b\}$ as $p = \text{Softmax}(w + \mathcal{N}(0, 1))$. We then modify $q(y_s)$ to control whether $y_s$ is smoothed:

**Smoothed** (Figure 5b): $q(y_s)$ is left unchanged. Since $m \ll n$, the probability that $y_s$ is smoothed is approximately $(n - m)/n$, which is close to 1.

**Smoothed with probability** $\approx 0.5$ (Figure 5c): We multiply $q(y_s)$ by 4.5, which empirically results in $y_s$ being smoothed with probability close to 0.5 for $m = 10$ and $n = 32000$.

**Not smoothed** (Figure 5d): We set $q(y_s) = \max_y q(y)$, making $y_s$ among the highest relative information-gain tokens and therefore very unlikely to be smoothed.

As shown in Figure 5, CP-$\Delta_r$ consistently achieves smaller bounds than CP-$\Delta$, and SCP-$\Delta_r$ further improves upon CP-$\Delta_r$. Moreover, the improvement of SCP-$\Delta_r$ becomes more pronounced as the likelihood of $y_s$ being smoothed increases, in accordance with Proposition 4.8. Error bars denote standard deviation (not standard error); results are averaged over 200 trials, and all differences between SCP-$\Delta_r$ and CP-$\Delta_r$ are statistically significant ($p < 10^{-4}$).

*Table 5.* Canary attack results. Best values are in bold. Underlined values are within two standard deviations of the random-canary baseline (i.e., not significantly above chance). Random-canary baseline: exposure computed for canaries of the same format that were not inserted into training.

| | REP $= 1$ | | REP $= 3$ | | REP $= 10$ | |
| --- | --- | --- | --- | --- | --- | --- |
| | Mean $\downarrow$ | 95% $\downarrow$ | Mean $\downarrow$ | 95% $\downarrow$ | Mean $\downarrow$ | 95% $\downarrow$ |
| No CP | 24 | 28 | 28 | 31 | 29 | 33 |
| CP-$\Delta$ | 5.2 | 13 | 8.8 | 17 | 11 | 20 |
| CP-$\Delta_{KL}$ | 16 | 25 | 23 | 33 | 25 | 33 |
| CP-Fuse | 2.7 | 5.5 | 3.8 | 7.1 | 4.2 | 7.2 |
| CP-$\Delta_r$ | 2.6 | 6.4 | 3.6 | 8 | 3.8 | 8.9 |
| SCP-$\Delta$-c | **1.5** | **4.4** | 1.6 | 3.4 | 1.8 | 4.1 |
| SCP-$\Delta$-b | 1.7 | 4.8 | **1.4** | **3.2** | **1.6** | **3.7** |

*Table 6.* PII average extraction length results with LoRA fine-tuning.

|  | AEL |
|---|---|
| No CP | $9.79 \pm 0.08$ |
| CP-$\Delta$ | $2.6 \pm 0.8$ |
| CP-$\Delta_{KL}$ | $5.0 \pm 1.3$ |
| CP-Fuse | $5.0 \pm 1.3$ |
| CP-$\Delta_r$ | $0.9 \pm 0.23$ |
| SCP-$\Delta$-c | $\underline{0.81} \pm 0.07$ |
| SCP-$\Delta$-b | $\mathbf{0.39} \pm 0.05$ |

### D.5. The Canary Method

Table 5 presents the full canary attack results on the *MathAbstracts* dataset. Results are qualitatively similar across different numbers of canary repetitions, with higher repetition leading to increased memorization for all methods except for both variations of SCP-$\Delta_r$. Notably, both variations of SCP-$\Delta_r$ remain within two standard deviations of the random canary baseline in 5 out of 6 evaluated settings.

### D.6. PII Attacks on LoRA Fine-Tuned LLMs

Table 6 reports the average extracted length (AEL) for the PII attack on the *Names2IDs* dataset when fine-tuning with LoRA for 20 epochs. Results are averaged over five trials with different training examples. Notably, LoRA did not prevent memorization, and these results are qualitatively similar to those obtained under non-LoRA fine-tuning (Table 2).

### D.7. Reasoning and Instruction-Following Evaluation

In addition to our empirical next-token accuracy evaluation and the KL-divergence bounds introduced by smoothing (Lemma A.2), we evaluate the impact of smoothing on downstream capabilities. We measure reasoning performance on MMLU (5-shot) (Hendrycks et al., 2021) and instruction-following performance on IFEval (Zhou et al., 2023).

For the MMLU evaluation, we use Llama-2-7b, while for the IFEval evaluation we use Llama-2-7b-chat. In both experiments, we set the smoothing parameter to $m = 10$, as in our main experiments. Table 7 reports the results.

The results are consistent with our next-token accuracy findings, demonstrating that smoothing preserves model utility.

*Table 7.* Utility evaluation with and without smoothing.

| Configuration | MMLU Accuracy (5-shot) | IFEval Prompt-Level Accuracy |
|---|---|---|
| No Smoothing | 45.83% | 31.42% |
| With Smoothing | 45.75% | 31.61% |

