# OpenReview forum: "Provably Protecting Fine-Tuned LLMs from Training Data Extraction while Preserving Utility"
_ICML.cc/2026/Conference — ICML 2026 regular_

### Official Review · Reviewer_745f · 2026-03-09

**Soundness:** 4
**Presentation:** 4
**Significance:** 3
**Originality:** 3
**Overall Recommendation:** 5
**Confidence:** 4

**Summary:**

The paper proposes SCP-$\Delta_r$, a defense mechanism based on Near Access Freeness (NAF, Vyas et al., 2023) designed to protect fine-tuned Large Language Models (LLMs) against Training Data Extraction (TDE) attacks. The algorithm leverages the observation that fine-tuning induces "Sparse Relative Probability Shifts" (SpaRPS), meaning significant probability changes are concentrated on a small subset of tokens. By operating on relative probability distributions and smoothing low-impact tokens using a base model, SCP-$\Delta_r$ preserves essential deviations while aggressively mitigating the probability spikes associated with memorization. Theoretically, the authors prove that SCP-$\Delta_r$ achieves orders-of-magnitude smaller NAF bounds than existing CP-style methods, which translates into strict mathematical limits on the information gain available to an adversary utilizing optimal likelihood-ratio attacks. Empirically, evaluations across multiple datasets demonstrate that SCP-$\Delta_r$ provides the strongest defense against various extraction methods—such as canary insertion, PII extraction, and token-by-token extraction—without exhibiting observable degradation in downstream model utility compared to unprotected baselines

**Compliance With Llm Reviewing Policy:**

Affirmed.

**Final Justification:**

The rebuttal and paper are consistent, I keep my score.

**Key Questions For Authors:**

I don't have any question

**Limitations:**

yes

**Strengths And Weaknesses:**

I am very supportive of this line of work: Fine-tuning LLMs on private, high-value data (e.g., enterprise data, medical records) is a fundamental important problem in our field.

The paper makes a rigorous connection between the Near Access Freeness (NAF) framework and Training Data Extraction (TDE). I think this connection is neat and original. By formulating bounds on the likelihood-ratio attack (LiRA), the authors provide mathematical guarantees limiting an adversary's information gain, which is a step up from purely empirical defenses.
On the other hand, the theoretical guarantees (Theorem 4.4 and 4.5) rely heavily on the assumption that the adversary utilizes a likelihood-ratio strategy (free-form token-by-token generation violates some of these strict optimality assumptions).

It is nice to see that the Sparse Relative Probability Shift (SpaRPS) property is empirically validated and so motivates the theory and the rest of the paper. Recognizing that memorization manifests as isolated probability spikes rather than global shifts justifies the methodological pivot to relative probabilities and smoothing.

The authors evaluate the method against a good set of baselines (no defense, CP-$\Delta$, CP-$\Delta_{KL}$, CP-Fuse) across diverse datasets (code, reasoning, PII, and synthetic canaries) and multiple attack vectors (adaptive and non-adaptive).
Maybe a limitation is that the NAF framework requires partitioning the dataset and training multiple constituent models. While inference is parallelizable, the training cost scales with the number of constituents. It seems that this could be a serious potential limitation for scalability.

---

> ### Author Rebuttal · Authors · 2026-03-27
>
> We thank the reviewer for the positive assessment of our work, including the theoretical analysis, the motivation provided by the SpaRPS property, and the comprehensive empirical evaluation across baselines, datasets, and attacks.
>
> **Training cost scaling with multiple constituent models**
>
> Regarding training cost, training multiple constituent models does not substantially increase total compute under a fixed data budget, as each model is trained on a partition of the data and training cost scales approximately linearly with the number of tokens.
>
> **Reliance on likelihood-ratio**
>
> Regarding the assumption of the likelihood-ratio attack (LiRA) usage by the adversary, Appendix C.1-C.2 discusses the conditions under which LiRA is optimal and shows that it remains a strong attack even when these conditions are relaxed. Importantly, our empirical evaluation includes extraction attacks beyond the strict LiRA optimality setting, demonstrating robustness in more general scenarios.

---

### Official Review · Reviewer_soF9 · 2026-03-13

**Soundness:** 3
**Presentation:** 3
**Significance:** 2
**Originality:** 3
**Overall Recommendation:** 5
**Confidence:** 3

**Summary:**

This paper introduces the SpaRPS property, which captures the sparsity of fine-tuning–induced relative probability shifts. Building on this property, the authors refine CP-Delta_r by incorporating base-model smoothing to propose SCP-Delta_r, and derive smaller NAF bounds to provide stronger guarantees for preventing data extraction.

**Compliance With Llm Reviewing Policy:**

Affirmed.

**Final Justification:**

The rebuttal addressed my concerns. I keep my score (5 Accept).

**Key Questions For Authors:**

1. The proposed method requires training two models on two disjoint data partitions. Does this effectively double the fine-tuning cost?

2. How well does the proposed method scale with respect to both the size of the fine-tuning dataset and the model size?

3. In Line 231, it appears that the exact values of \tilde{p} and \tilde{q} are not required. Is that correct?

**Limitations:**

Yes

**Strengths And Weaknesses:**

**Soundness**

On the theoretical side, the motivation and formulation of the SpaRPS property and SCP-Delta_r are clearly explained. The relationship between SCP-Delta_r and the LiRA bound is well established. In addition, the authors empirically validate the tightness of the newly derived bounds.

On the experimental side, the paper demonstrates improved performance in protecting fine-tuned models from data extraction compared to other NAF-based methods, while still maintaining strong model utility.
However, this paper lacks a large-scale evaluation of the proposed method, including models of different sizes, and more non-NAF-based methods.

**Presentation**

The paper is generally well presented and easy to follow.


**Significance**

While empirical mitigation methods for data extraction often lack theoretical guarantees, and differential privacy–based approaches typically sacrifice model utility, the proposed NAF-based method provides both theoretical analysis and strong empirical performance. In particular, it achieves a promising balance between privacy protection and model utility. One concern is that the proposed method at least doubles the fine-tuning cost. Generally, this work is therefore valuable for the research community studying privacy-preserving fine-tuning of large language models.


**Originality**

The proposed SpaRPS property and SCP-Delta_r formulation are novel contributions.

---

> ### Author Rebuttal · Authors · 2026-03-27
>
> We thank the reviewer for the positive assessment of our motivation, theoretical analysis, empirical results, and the originality of SpaRPS and SCP-$\Delta_r$.
> Regarding non-NAF based methods, in this work, we focus on improving and evaluating NAF-based methods under strong training data extraction attacks. Prior works have shown that NAF-based methods outperform non-NAF approaches in various settings (Vyas et al., 2023; Abad et al., 2025; Segal et al., 2025).
> We would like to emphasize that, beyond SpaRPS and SCP-$\Delta_r$, our work demonstrates that NAF-based methods can be highly susceptible to TDE attacks, a finding we consider a significant contribution.
>
> **Answers to questions**
>
> 1. Training two constituent models does not increase the total compute under a fixed data budget, as each model is trained on half of the data and training cost scales approximately linearly with the number of training tokens. Thus, the overall training cost is comparable to standard fine-tuning. Furthermore, in a distributed environment, training the two models, each on half the data, could actually be faster in practice as it can be done in parallel. We will clarify this in the final version.
> 2. While we do not vary model or dataset size in our experiments, our theoretical analysis is independent of these parameters. Additionally, prior NAF-based work has demonstrated effectiveness across a range of model sizes and dataset sizes (Vyas et al., 2023; Segal et al., 2025), suggesting that the approach is not tied to a specific scale. Exploring this more systematically is an interesting direction for future work.
> 3. It is correct that the values of $\tilde{p}$ and $\tilde{q}$ in line 231 are not required during execution. The statement in Line 231 is intended to show that SCP-$\Delta_r$ can be viewed as CP-$\Delta_r$ applied to specific distributions, allowing the theoretical analysis to carry over. We will clarify this in the paper.

---

> > ### Author Rebuttal · Reviewer_soF9 · 2026-04-04
> >
> > Thanks for your clarifications. I'll keep my score.

---

### Official Review · Reviewer_ucmo · 2026-03-13

**Soundness:** 2
**Presentation:** 3
**Significance:** 3
**Originality:** 3
**Overall Recommendation:** 4
**Confidence:** 3

**Summary:**

This paper studies defenses against training data extraction attacks in fine-tuned models. The authors observe that fine-tuning often introduces sparse probability spikes corresponding to memorized training data and propose a smoothing-based defense that suppresses these shifts by mixing the fine-tuned distribution with that of a base model. The proposed method has theoretical analysis and is evaluated against several extraction attacks.

**Compliance With Llm Reviewing Policy:**

Affirmed.

**Final Justification:**

It deserves weak accept considering the efforts made by the authors.

**Key Questions For Authors:**

Please see the weakness.

**Limitations:**

The main limitation of the paper is that the core assumption is not well judged. The proposed defense critically relies on the assumption that fine-tuning introduces sparse probability shifts, but the paper does not provide strong empirical evidence that this property holds across real fine-tunings.

**Strengths And Weaknesses:**

Strengths
1. The paper provides an intuitive and interesting view that memorization in fine-tuned models manifests as sparse probability spikes, which motivates the follwing defense.
2. The method operates at the probability level and does not require retraining the model, making it relatively easy to integrate into existing systems.

Weaknesses
1. The proposed method assumes that fine-tuning only introduces sparse probability shifts. This assumption may not hold in many practical fine-tuning scenarios (e.g., domain adaptation), where the output distribution can change broadly.
2. The defense depends on the availability of a reference base model. In some real deployments, the original base model may not be accessible.
3. The theoretical analysis assumes a partitioned training setting where sensitive examples do not appear in at least one constituent model. This assumption is unlikely to hold in real-world fine-tuning pipelines.
4. The utility is primarily measured using next-token accuracy. It remains unclear how the proposed smoothing affects long-form generation, reasoning tasks, or other realistic downstream usecases.
5. Since the method mainly reduces probability changes, memorized tokens may still remain relatively more likely than alternatives. Therefore,  the system may still be potentially vulnerable to stronger adaptive attacks.

---

> ### Author Rebuttal · Authors · 2026-03-27
>
> We thank the reviewer for acknowledging our work's motivation and the easy-to-integrate nature of our approach. We highlight two additional contributions of our work mentioned in the Introduction:
> 1. We link NAF to training data extraction (TDE) protection through rigorous theoretical analysis.
> 2. We show that existing NAF-based approaches are vulnerable to training data extraction, both conceptually and in practice.
>
> **Responses to weaknesses**
>
> 1. Regarding the claim that our method assumes fine-tuning introduces sparse probability shifts, we clarify that this is not the case. Our method **does not assume** the model is naturally sparse; it explicitly **enforces** sparsity to guarantee privacy (as stated in the introduction and in Section 4). In fact, as discussed in the first paragraph of Section 4, Figure 1a shows that sparsity does not naturally hold in the corresponding experiment (e.g., the parameter $B$ is not close to 0, as would be expected if sparsity held), which motivates our sparsity enforcing smoothing technique. We will revise the text to make this distinction between empirical observation and algorithmic enforcement clearer.
> 2. Regarding the reliance on a reference model, we note that while our method relies on a reference distribution, it need not be the original base model. In particular, the SCP-$\Delta_r$-c variant (Section 5) uses a constant logit vector as a reference model. Thus, our approach can be applied even when the original base model is unavailable.
> 3. Regarding the concern of the likeliness of the partition assumption, we note that our guarantees rely on the standard NAF requirement that at least one partition does not contain the sensitive example. We agree this requirement may not always hold naturally in practice, which we explicitly discuss as a limitation. We will add relevant future work directions to address this, such as grouping similar documents to the same partition.
> 4. Regarding the concern about our utility evaluation, we note that CP-style methods are not expected to degrade utility, since for non-sensitive inputs the output distributions of the constituent models are similar (Vyas et al., 2023; Abad
> et al., 2025). Accordingly, we use next-token accuracy as a validation that smoothing preserves this property, rather than as a comprehensive evaluation of downstream performance. We also provide theoretical guarantees in Appendix A, where we derive bounds on the potential utility degradation of our method. We will clarify this in the final version of the paper.
> 5. As for the claim that our method mainly reduces probability changes, we note that our method suppresses probability spikes by replacing them with values aligned with a safe reference distribution, making the resulting probabilities largely insensitive to the magnitude of the original spike (or the original probability change). This intuition is supported by our theoretical bounds, which remain small even in the presence of large spikes (Remark 4.6).
>
> **Limitations**
>
> We discuss several limitations in the corresponding section (e.g., the partition structure). Regarding the main limitation identified - the sparsity of probability shifts - please see our response to Weakness 1. To summarize: our method does not assume natural sparsity; rather, it enforces it through base-model smoothing to guarantee privacy. As explicitly shown in Figure 1a, we recognize that sparsity does not naturally hold in practice (as $B$ is not close to 0), which serves as the primary motivation for our sparsity-enforcing mechanism.

---

> > ### Author Rebuttal · Reviewer_ucmo · 2026-04-01
> >
> > Thank you for the rebuttal. The response clarifies several details, but my main concerns still remain.
> >
> > 1. The response states that the method enforces sparsity rather than assuming natural sparsity. While this point explains the design, it does not fully address whether such enforced sparsity is realistic in real-world fine-tuning settings where probability shifts may be widespread.
> >
> > 2. The theoretical guarantees still rely on the partition assumption, which may not hold in many real-world fine-tuning pipelines.
> >
> > 3. The utility evaluation is mainly based on next-token accuracy, which is not sufficient to show that performance is preserved in more realistic tasks such as long-form generation or reasoning.
> >
> > 4.  The robustness to stronger adaptive attacks is not convincing,  as the rebuttal only provides intuition but limited empirical evidence.

---

> > > ### Author Response · Authors · 2026-04-03
> > >
> > > Thank you for the rebuttal acknowledgment.
> > >
> > > **Responses to remaining concerns**
> > > 1.  **Enforced Sparsity:** We clarify that the distribution produced by the smoothing stage is SpaRPS (Definition 3.2) as per Remark 4.1, which is proven in Appendix B.3. This holds true regardless of the input distributions and for any fine-tuning setting. Consequently, our theoretical guarantees remain valid and provide protection even when probability shifts are widespread, as the smoothing process transforms them into sparse shifts. Furthermore, we validate both the protection and utility of our method across a diverse range of datasets: synthesized PII, mathematics papers, code generation, and story writing. We consider these datasets to be representative of realistic, real-world fine-tuning scenarios.
> > > 2.  **Partition Assumption:** We agree that the partition assumption is a limitation, as noted in the paper. We believe this could be addressed using a preprocessing step that partitions the data strategically; however, such a step is beyond the scope of this work. It is customary for NAF-based research to treat this as an independent preprocessing step that is addressed externally, depending on the specific scenario (Vyas et al., 2023; Segal et al., 2025; Abad et al., 2025).
> > > 3.  **Utility Evaluation:** In addition to our empirical next-token accuracy evaluation, we also bound the KL-divergence introduced by smoothing (Lemma A.2). Nevertheless, to fully address the concern, we evaluated reasoning capabilities via MMLU (5-shot) and instruction-following/long-form generation via IFEval. We used Llama-2-7b for the MMLU experiment and Llama-2-7b-chat for the IFEval experiment, with the smoothing parameter set to $m=10$ (consistent with our main experiments). As shown below, the results align with our next-token accuracy findings, demonstrating comparable utility with and without smoothing. We will include these results in the final paper.
> > >   | Configuration | MMLU Accuracy (5-shot) | IFEval (Prompt-level Accuracy) |
> > >   | :--- | :---: | :---: |
> > >   | **No Smoothing** | 45.83% | 31.42% |
> > >   | **With Smoothing** | 45.75% | 31.61% |
> > > 4. **Robustness to Stronger Adaptive Attacks:** We clarify that our evaluated attacks are strong adaptive attacks. We are unaware of more powerful attacks within the black-box scenario. Our theoretical analysis bounds the effectiveness of the Log-Likelihood Ratio Attack (LiRA), which is the **optimal attack** an adversary could utilize under certain assumptions (Subsection 4.2). Even when these assumptions do not hold fully, LiRA typically remains relevant, as discussed in Appendix C.1–C.2. Due to this optimality argument, we are unaware of a method to further strengthen our attacks and are thus unable to provide empirical evidence on handling "stronger" attacks. Further research is required to determine if attacks stronger than LiRA exist under the black-box scenario with a non-restricted adversary. We note that when the adversary is restricted, LiRA might not be optimal [1, 2], which occurs because the attack is weakened in that context. However, our empirical attacks and theoretical analysis assume a non-restricted adversary with access to training data prefixes and a perfect reference model (see Appendix C.2 on interpreting our attacks as using a reference model). To the best of our knowledge, this implies our attacks are as strong as any known attack can be in the black-box setting. We will clarify this optimality argument in the paper.
> > >
> > > **References**
> > >
> > >  [1] Fu et al. (2023). Practical membership inference attacks against fine-tuned large language models via self-prompt calibration.
> > >
> > > [2] Zhang et al. (2025). Min-K%++: Improved Baseline for Pre-Training Data Detection from Large Language Models.

---

### Official Review · Reviewer_Yi1D · 2026-03-16

**Soundness:** 3
**Presentation:** 3
**Significance:** 3
**Originality:** 2
**Overall Recommendation:** 4
**Confidence:** 4

**Summary:**

The paper proposes a scheme for preventing memorization in finetuning that relies on NAF-like properties. It adapts an idea from the access control applications of these schemes around using relative probability distributions rather than absolute probability distributions, which dramatically improves the approach, and adds a smoothing idea to improve performance further. The paper also proves theorems bounding membership inference attack success using these types of schemes. Overall, their approach works the best at protecting against training data extraction while maintaining good utility.

**Compliance With Llm Reviewing Policy:**

Affirmed.

**Final Justification:**

I am satisfied with the rebuttal. I think the paper has a solid technical contribution and no major soundness issues.

**Key Questions For Authors:**

I don't have any questions I think would significantly change my opinion of the paper, but I welcome the authors correcting misunderstandings of the paper if they exist in my review!

**Limitations:**

The paper discusses some limitations.

**Strengths And Weaknesses:**

Soundness:
The experiments and theorems in the paper appear sound. The fundamental ideas in the paper are sound. The privacy attacks used in the paper are a good set and convince me.

Presentation:
Overall, the paper is quite well presented. The paper is well structured and the ideas and experiments are presented clearly.

I'm not sure "CP" is ever written unabbreviated in the paper. The different variants are also a bit tough to manage.

Significance:
The topic considered is important. The approach taken is a good one and could influence future work.

The biggest limitation in terms of how likely this will influence applications is in the problem of duplication. Deduplication is ruled out-of-scope here, but a huge reason why LLMs memorize is because of near-duplicates in the training data. These types of schemes do not grapple with this issue at all - duplicates will likely be distributed between partitions of the dataset evenly, meaning that both models could end up memorizing likely-to-be-duplicated examples anyways. One way of protecting against this is by attempting to group similar documents into the same partition, but note that this is just as hard as deduplication! This is briefly discussed in the Discussion section, but I think it's quite important and would encourage experiments in this direction if the authors want to justify this isn't a big limitation.

Originality:
This is the biggest weakness of the paper in my opinion. The idea of relative probability in these kinds of copyright protection schemes is a clearly good one, but I'd say the relative probability idea being used for memorization is about 80% of what I find nice about this paper, but this is just repurposing existing machinery to a (very) close area.

The rest of the paper beyond this is quite nice in terms of originality. The smoothing idea is a small but nice tweak, and the theorems around MIAs are quite nice given the similarity of these NAF schemes and MIAs - I'm surprised nobody has analyzed this before (I mean this as a compliment!).

---

> ### Author Rebuttal · Authors · 2026-03-27
>
> We thank the reviewer for acknowledging the soundness, presentation and significance of our work and the originality of our smoothing idea and our theorems linking NAF and MIA. We note that our theoretical analysis is rigorous and involves connecting NAF theory and relative probabilities to membership inference and optimal training data extraction attacks via the likelihood ratio attack (LiRA), which, as the reviewer noted, has not been previously explored to our knowledge.
>
> **Originality of our work**
>
> While relative probabilities were previously explored in access-control settings, their application to TDE is non-trivial: we show how relative probability bounds translate into guarantees against likelihood-ratio-based extraction attacks, and that standard NAF guarantees can be insufficient in this setting. Furthermore, our smoothing technique is the necessary bridge that makes relative probabilities work for TDE. Without smoothing, the theoretical bounds remain empirically large, especially for rare tokens, for which protection is particularly critical (see Figure 2a and the canary and PII results in Table 2).
> To further clarify the originality of our work beyond the smoothing technique and our theoretical analysis, we emphasize that our contributions are not limited to a new method, but also provide novel insights and understanding of LLMs fine-tuning and NAF-based approaches. In particular, we highlight two additional contributions mentioned in the introduction section:
> 1. We introduce the SpaRPS property (Section 3), which gives insights into how probability distributions change during LLM fine-tuning and may be useful beyond LLM security.
> 2. We show that previously proposed NAF-based approaches are vulnerable to several types of training data extraction attacks (Table 2), deepening the community’s understanding of these methods.
>
> **Duplication and deduplication**
>
> Thank you for your comment on duplication and grouping similar documents. As discussed in our limitation section, our guarantees rely on the standard NAF requirement that at least one partition does not contain the sensitive example (or a close variant). Because of that, duplication poses a challenge for our method, as you mentioned. We view grouping similar documents into the same partition as a promising direction for future work and will mention this in the final version. We note that deduplication is a complementary but fundamentally different approach, as it does not provide theoretical guarantees and requires high-precision identification to avoid harming utility, whereas grouping for partitioning can tolerate noisier similarity signals since no data is removed.

---

> > ### Author Rebuttal · Reviewer_Yi1D · 2026-04-05
> >
> > I thank the authors for clarifications raised in the rebuttal.

---

### Decision · Program_Chairs · 2026-04-30

**Decision:**

Accept (regular)

**Comment:**

The paper focuses on addressing a basic and important problem: how to protect sensitive data that is used for (not training but) fine-tuning LLMs. The paper uses ideas regarding relative probability theory and uses a so-called Near Access Freeness (NAF)-based algorithm that smoothest certain conditional token sampling probabilities while not affecting the utility of the model. Technically, the approach relies on partitioning the data into two sets, one of which does not have sensitive information. This is an extra burden that shall be taken, and there are also theoretical results proved about the resilience of the approach based on this data partitioning assumption. Overall, the reviewers believe that the problem, the approach, and experiments are important and convincing, and hence I recommend the paper to be accepted.